
# Investigation of space-borne trace gas products over St. Petersburg and Yekaterinburg, Russia by using COCCON observations

Carlos Alberti[1*], Qiansi Tu[1*], Frank Hase[1], Maria V. Makarova[2], Konstantin Gribanov[3], Stefani C. Foka[2], Vyacheslav Zakharov[3], Thomas Blumenstock[1], Michael Buchwitz[4], Christopher Diekmann[1], Benjamin Ertl[5], Matthias M. Frey[1,a], Hamud Kh. Imhasin[2], Dmitry V. Ionov[2], Farahnaz Khosrawi[1], Sergey I. Osipov[2], Maximilian Reuter[4], Matthias Schneider[1], and Thorsten Warneke[4]

[1]Institute of Meteorology and Climate Research (IMK-ASF), Karlsruhe Institute of Technology, Karlsruhe, Germany
[a] now at National Institute for Environmental Studies (NIES), Tsukuba, Japan
[2]Department of Atmospheric Physics, Faculty of Physics, St. Petersburg State University, St. Petersburg, Russia
[3]Institute of Natural Sciences and Mathematics, Ural Federal University, Yekaterinburg, 620000, Russia
[4]Institute of Environmental Physics and Institute of Remote Sensing, University of Bremen, Bremen, Germany
[5]Karlsruhe Institute of Technology, Steinbuch Centre for Computing (SCC), Karlsruhe, Germany
*These authors contributed equally to this work.

*Correspondence to*: Carlos Alberti (carlos.alberti@kit.edu) and Qiansi Tu (Qiansi.tu@kit.edu)

**Abstract.** This work employs ground- and space-based observations, together with model data to study columnar abundances of atmospheric trace gases ($XH_2O$, $XCO_2$, $XCH_4$, and $XCO$) in two high-latitude Russian cities, St. Petersburg and Yekaterinburg. Two portable COllaborative Column Carbon Observing Network (COCCON) spectrometers were used for continuous measurements at these locations during 2019 and 2020. Additionally, a subset of data of special interest (a strong gradient in $XCH_4$ and $XCO$ was detected) collected in the framework of a mobile city campaign performed in 2019 using both instruments is investigated. All studied satellite products (TROPOMI, OCO-2, GOSAT, MUSICA IASI) show generally good agreement with COCCON observations. Satellite and ground-based observations at high latitude are much sparser than at low or mid latitude, which makes direct coincident comparisons between remote-sensing observations more difficult. Therefore, a method of scaling continuous CAMS model data to the ground-based observations is developed and used for creating virtual COCCON observations. These adjusted CAMS data are then used for satellite validation, showing good agreement in both Peterhof and Yekaterinburg cities. The gradients between the two study sites ($\Delta Xgas$) are similar between CAMS and CAMS-COCCON data sets, indicating that the model gradients are in agreement with the gradients observed by COCCON. This is further supported by a few simultaneous COCCON and satellite $\Delta Xgas$ measurements, which also agree with the model gradient. With respect to the city campaign observations recorded in St. Petersburg, the downwind COCCON station measured obvious enhancements for both $XCH_4$ (10.6 ppb) and $XCO$ (9.5 ppb), which is nicely reflected by TROPOMI observations, which detect city-scale gradients of the order 9.4 ppb for $XCH_4$ and 12.5 ppb $XCO$, respectively.



## 1 Introduction

Since human beings exist on the Earth's surface, their activities have deteriorated the environment in several manners. The increase of the global population, the globalization of the economy, the growing industry and the transport sector are only some of the most important causes, which has increased the anthropogenic emission. These activities require the use of huge amount of energy, among which the fossil fuels such as coal, oil and natural gas are the main sources since the industrial era. Global warming is one of the most discussed negative effects caused by the anthropogenic emissions of GHGs. The effect is caused by the anthropogenic emissions of greenhouse gases (GHGs), mainly carbon dioxide ($CO_2$), methane ($CH_4$), and nitrous oxide ($N_2O$). These gases absorb part of the infrared emission of the Earth, corresponding to their molecular structure. Consequently, the Earth's surface temperature increases, resulting in melting of glaciers and the Greenland and Antarctic ice sheets, sea level rise, droughts, and other negative effects. Global warming leads to a climate change which, in turn, leads to a disruption in the hydrological cycle, resulting in unpredictable weather patterns. Therefore, huge efforts are needed on all levels: local, national and global are required in order to slow down the GHGs emission tendency. Such efforts require not only a panel of scientists and engineers but also politics and policy/decision makers for implementing effective measures. On that regard countries have debated since more than three decades, and such meetings produced several important agreements. In 1992, the first global deal that focused on climate change was created: the UN Framework Convention on Climate Change (UNFCCC), which established the annual Conference of the Parties (COP). Based on this meeting the Kyoto Protocol and the Paris Agreement were created. The first one began on 2005 and its main aim was committing industrialized economies to reduce the emission of GHGs for defined and agreed targets. Unfortunately, after more than one decade the global anthropogenic emissions of GHGs continued increasing (Harris et al., 2012). The second one came into force on November 2016, which aims to limit the global warming below 2 °C or even below 1.5 ° C. Such objective can be only possible through reducing the GHGs emitted into the atmosphere. Although the majority of cities have enacted initiatives to measure and control pollution, the majority of developed interventions are localized (Miller et al., 2013; Seinfeld and Pandis, 2016). In general, the governments of most countries globally have failed to enact effective measures of addressing anthropogenic pollution (Roger et al., 2016).

In summary, we need to know more about the natural sources and sinks of GHGs into the atmosphere to better understand climate change, which will in turn allow better projections of their future under climate change conditions. Additionally, we need to monitor the anthropogenic emissions, e.g., in the context of the Paris Agreement. Because $CO_2$, which is the most important GHG, is long lived. Both applications require to measure relatively small changes over a large background concentration and this is only possible with high accuracy and state-of-the-art instrumentation, which nowadays has become more crucial than ever. On that framework, national and international consortiums and agencies have been measuring GHGs in the atmosphere with different sampling methods, and different spatial-vertical resolutions and accuracies. Remote sensing is one of the approaches through which GHGs can be continuously measured on a global scale. Such measurements can be made with space-based techniques by using satellites, like the SCanning Imaging Absorption spectroMeter for Atmospheric





CartograpHY (SCIAMACHY), Greenhouse Gases Observing Satellite (GOSAT), Orbiting Carbon Observatory-2 (OCO-2), and Troposheric Monitoring Instrument (TROPOMI) onboard of Sentinel-5 Precursor (S5P). For the validation of data products from these space-borne sensors, remote sensing observations are performed by ground-based networks: the NDACC FTIR (Fourier Transform InfraRed) network (https://www.ndaccdemo.org/, last access 11 Jul. 21) and the Total Column Carbon Observing Network (TCCON) (http://www.tccon.caltech.edu/, last access 11 Jul. 21) which is regarded as the reference

network for columnar GHG measurements, recently supplemented by the COllaborative Column Carbon Observing Network (COCCON) (Frey et al., 2019). Current constellation of satellites provides highly accurate results with a global coverage, nevertheless for these and future GHG satellite missions the aforementioned highly accurate ground-based FTIR measurements are crucial for their calibration and validation. The TCCON network has been established since 2004. However, the expensive instrumentation and required maintenance effort limits the number of stations.  Recently, TCCON has been complemented by

the COCCON network, which uses low-resolution Bruker EM27/SUN FTIR spectrometers (in the following referred to as COCCON instrument), developed by Karlsruhe Institute of Technology (KIT) in collaboration with Bruker (Gisi et al., 2012; Hase et al., 2016). This instrument is a portable unit and easy to operate for non-experts. It has been shown in several peer-review studies that COCCON instruments enable to retrieve GHGs with high precision and accuracy, and several campaigns have been carried out even at remote sites.

The EU project VERIFY (https://verify.lsce.ipsl.fr/, last access 2 July, 2021) aims to quantify/estimate the anthropogenic and natural GHG emissions based on atmospheric measurements, emission inventories and ecosystem data. Within this project two cities in Russia (St. Petersburg and Yekaterinburg) were selected with the objective of improving our understanding of a key important region with anticipated huge biosphere fluxes and potentially extensive carbon sinks (Reuter et al., 2014). Because only a few measurements are available on this region, two different campaigns were carried out there in the framework

of VERIFY: continuous measurements at fixed locations in both places and also a mobile city campaign targeting at St. Petersburg emissions (Emission Monitoring Mobile Experiment, EMME). In the city campaign, two COCCON spectrometers were placed up- and downwind of St. Petersburg in 2019. With the obtained results, the emission ratios for the city emissions were quantified and compared with the bottom-up estimation as presented in Makarova et al. (2020). From these campaign data, the $CO_2$, $CH_4$, $NO_x$ and CO fluxes were estimated as well. Estimation of the anthropogenic $CO_2$ emissions using ODIAC

and the FTIR measurements are presented by Timofeyev et al. (2020), while the $CH_4$ emission intensities are presented by Foka et al. (2020). Additionally, the EMME campaign was extended in 2020 with only one spectrometer moved between the upwind and downwind sides. The integral $CO_2$ city emission for both periods are investigated by Ionov et al. (2021).

In contrast to the papers above, this paper focuses on the complete set of COCCON measurements collected in the framework of VERIFY to validate and inter-compare TROPOMI, OCO-2, GOSAT, MUSICA IASI and Copernicus Atmosphere

Monitoring Service (CAMS). Additionally, a scaling method is developed and its results are used to better inter-compare satellite products. This method is based on COCCON measurements at both sites to scale CAMS $XCO_2$, $XCH_4$ and XCO. The effectiveness of this method is proved by using different sub-sets of $XCH_4$ retrieved from the densest observations from the reference COCCON spectrometer (FTS#37) at Karlsruhe during the period of January 2018 – December 2020. Because GHGs



surface fluxes are imprinted in the atmospheric concentrations, in order to learn about them it is imperative to accurately

estimate their respective atmospheric gradients. On that regard, the gradients for $XCO_2$, $XCH_4$ and XCO are calculated between both studied cities during the shared measurement period. Finally, a city-scale transport event occurred during the city campaign and tracked by TROPOMI is presented in this study.

## 2 Russian Campaign location and set-up

Within the VERIFY project, two cities in Russia (St. Petersburg and Yekaterinburg) were chosen as target regions. The main

aim was to collect observations for evaluating $XCO_2$ gradients and the XCO / $XCO_2$ ratios in a very important region with high emissions and large biosphere fluxes in Eastern Europe. To achieve the foreseen objectives two different activities were carried out: a mobile city campaign (see section 2.2) and continuous measurements in two fixed locations: Peterhof (15 months) and Yekaterinburg (6 months) (see section 2.3).

### 2.1 Stability of the COCCON spectrometers during the campaign period

Measurements of very high precision and accuracy are required for correctly retrieving the columnar GHG abundances in the atmosphere. This can be well achieved with the portable FTIR spectrometers as the EM27/SUN spectrometer. For ensuring the optimum level of accuracy, prior to the campaign, the two instruments utilized in the campaign were checked, characterized and calibrated and the residual instrument-specific calibration factors of $XCO_2$, XCO, $XCH_4$ and $XH_2O$ with respect to the COCCON network reference were determined. For demonstrating the stability of the spectrometers, the calibration has been

redone after the campaign. This calibration work is described in Sect. 2.1.1 and 2.1.2.

### 2.1.1 Instrumental Line Shape (ILS) characterization

An important step in order to find any kinds of instrumental malfunction is the laboratory calibration. Open-path measurements described by Frey et al. (2015) are performed for recognizing channelling effects, increased noise levels, out-of-band artefacts, and for characterizing the instrumental line shape (ILS). The ILS for both instruments was determined at KIT before and after

the campaign in order to track their stability and thus, their performance. The ILS is given in terms of modulation efficiency (M. E.) and phase error (Table 1).

**Table 1: ILS in terms of modulation efficiency (M.E.) and phase error calculated before and after the campaign for instruments FTS#80 and FTS#84.**

| Instrument | Date | M. E. | Phase error |
|---|---|---|---|
| FTS#80 | 2018-04-17 | 0.9865 | -0.00275 |
|  | 2020-06-04 | 0.9861 | -0.01295 |
| FTS#84 | 2018-03-27 | 0.9900 | -0.00009 |
|  | 2020-06-04 | 0.9871 | 0.00083 |



### 2.1.2 Side-by-side measurements

After the instruments were calibrated, solar side-by-side measurements between the instruments used in the campaign (FTS#80 and FTS#84), the COCCON reference and the TCCON spectrometer operated at the same location were carried out at KIT. These measurements served to find the instrument-specific calibration factors for each retrieved gas. These factors are calculated with respect to the COCCON reference and help to harmonise the results for all COCCON spectrometers. Such measurements took place before (18 and 19 April, 2018) and during (12 April, 2019) the campaign. The later one served for

crosschecking whether the instruments kept the same behaviour and performance. These results can be seen in Figure 1 and Figure 2, respectively and the correction factors are listed in Table 2.

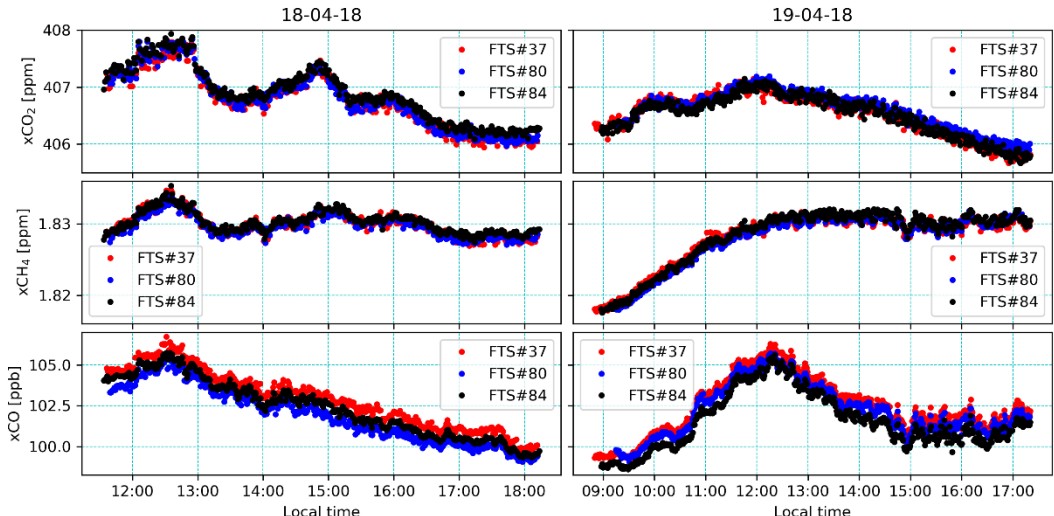

**Figure 1: Side-by-side measurements before the instruments were shipped to Russia. Comparisons between instrument no.1 (FTS#37), which is the COCCON reference unit operated at KIT, and instruments FTS#80 and FTS#84.**





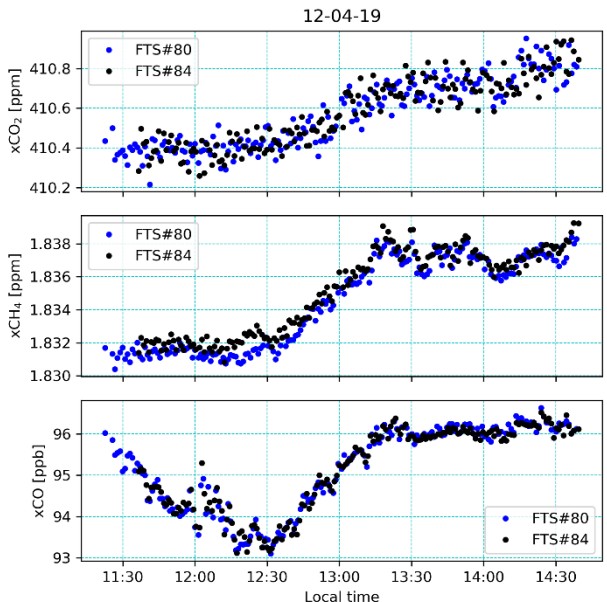


**Figure 2: Side-by-side measurements during the campaign but only with instruments FTS#80 and FTS#84.**

From the measurements shown in Figure 1, the correction factors for XCO$_2$, XCO and XCH$_4$ measured by the two instruments are calculated as described in Frey et al. (2019). These results are averaged and later used for scaling the results for each of the retrieved GHG analysed in this study as presented in Table 2.

**Table 2: Correction factors for instruments FTS#80 and FTS#84.**

| Instrument | Date | XCO$_2$ factor | XCH$_4$ factor | XCO factor |
|---|---|---|---|---|
| **FTS#80** | 18-19 April 2018 | 0.99988 | 1.00013 | 1.00636 |
| | 31 October 2020 | 0.99981 | 1.00042 | 1.00264 |
| | *Absolute drift* | 6.765e-05 | 2.966e-4 | 3.721e-3 |
| | ***Used value*** | ***0.99984*** | ***1.00028*** | ***1.00450*** |
| **FTS#84** | 18-19 April 2018 | 0.99990 | 0.99987 | 1.00748 |
| | 13 June 2021 | 0.99967 | 0.99953 | 1.00171 |
| | *Absolute drift* | 2.242e-4 | 3.333e-4 | 5.774e-3 |
| | ***Used value*** | ***0.99978*** | ***0.99970*** | ***1.00460*** |

**2.2 EMME campaign**

The EMME campaign is described in detail by Makarova et al. (2020), and here we summarize only the most relevant details of it. Because the aim of this campaign was to quantify the CO$_2$ emissions, CO/CO$_2$ emission ratios and the estimation of the CO$_2$, CH$_4$ and CO fluxes, two mobile COCCON FTIR spectrometers were used in order to retrieve the required GHG species.

Both instruments were located in the up- and downwind of the St. Petersburg city ring. This campaign was not made in a



continuous acquisition mode but the active phases were scheduled according to the weather forecast. The basic idea is to select the deployment position of each instrument one day before good meteorological conditions appeared. The wind forecast, and the orientation of the city's $NO_2$ plume as modelled by HYSPLIT were used as prediction tools and the positions of the COCCON spectrometers were selected accordingly. In addition, during a measuring day, the Russian partners carried out

mobile zenith DOAS measurements in order to measure the $NO_2$ total column flux over the city in a near real time manner. The second input helped to readjust the location of one or both spectrometers in case of deviations from the predicted plume orientation. Following this approach, a total of 11 successful measurement days were carried out during March to April 2019. An overview of the collected COCCON data is presented in Figure 3, from that figure is remarkable the enhancement on 25-04-19. This measurement day is presented as a plume transport event in a city-scale domain tracked by TROPOMI as

complement of the results shown by Makarova et al. (2020).

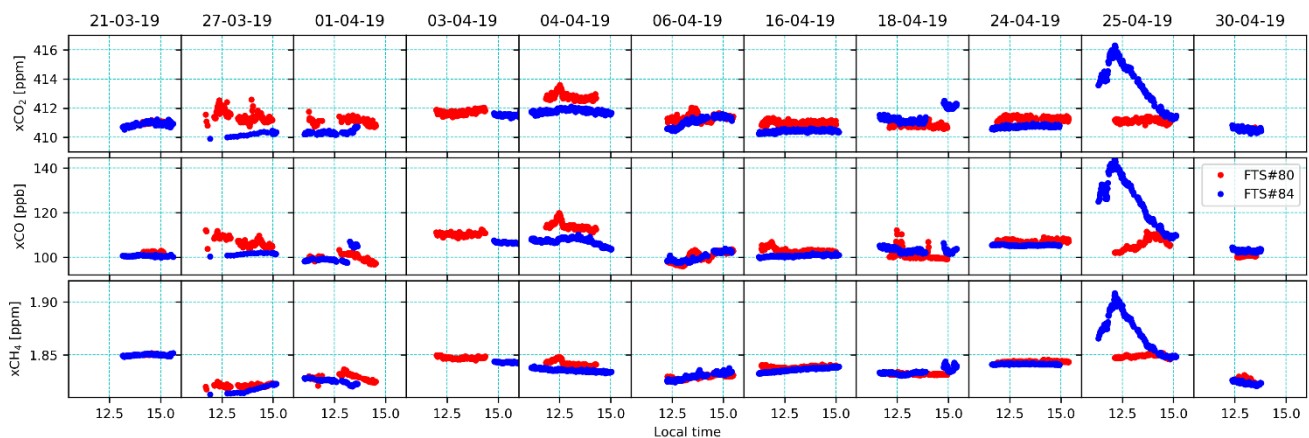

**Figure 3: General overview of the full campaign results collected with the COCCON spectrometers (Makarova et al., 2020).**

## 2.3 Ground-based FTIR measurements at Peterhof and Yekaterinburg

For the continuous, long-baseline campaign, the instrument FTS#80 remained at Peterhof station at the St. Petersburg State

University and continued operation there, while the other spectrometer FTS#84 was moved to Yekaterinburg.

### 2.3.1 Peterhof (59.88°N, 29.83°E)

Peterhof is a suburb of St. Petersburg located approximately 35 km southwest from St. Petersburg's city centre. The instrument in Peterhof was operated by the Russian partners at the Atmospheric Physics Department of the Faculty of Physics at St. Petersburg State University. The instrument was set up on every sunny day (out from the city campaign period) at the 2nd floor

of the FTIR remote sensing group (See Figure 4). Eighty-four measurement days were collected between January 2019 and March 2020 as it can be seen in Figure 5. From that figure, it is remarkable the larger XCO observed values on 06 August 2019 in comparison with all the other days. For more details, see Figure A- 1a and b where the spatial distribution of TROPOMI $XCH_4$ and XCO, and wind speed and direction, respectively for this day are presented. Additionally, Figure A- 1c shows the





time series for COCCON $XCO_2$, $XCH_4$ and XCO for that day and the enhancements are all observed in the three species. It
seems that these large values could be related to a plume transport from a heavily industrialized area coming from Lappeenranta
city, which is located in the southeast of Finland and approximately 160 km away from Peterhof. In order to confirm this,
Figure A- 2a shows that the yearly CO emissions came from the "Combustion from manufacturing sector" taken from EDGAR
V05 inventory (latest available: 2015), together with the backward trajectories calculated by using HYSPLIT (HYbrid Single-
Particle Lagrangian Integrated Trajectories) model (https://www.ready.noaa.gov/HYSPLIT_traj.php, last access: 04 August
2021) and arrived in Peterhof on that day (see Figure A- 2b). This confirms that the wind comes from the area where huge
anthropogenic CO sources are located. Another possibility could be an even closer local source, like a small fire.

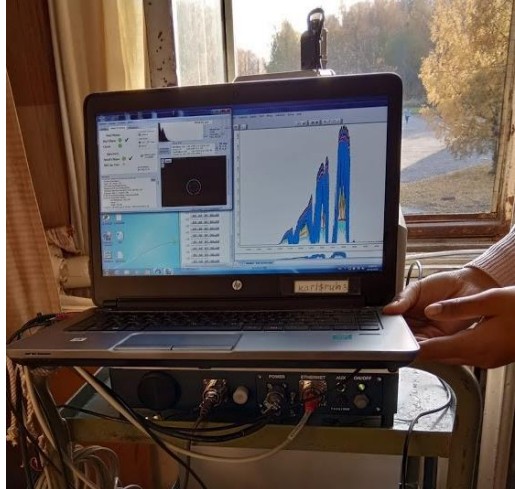

**Figure 4: Instrument setup at Peterhof. A huge window allowed measurements from ~10:00 to ~15:30 (local time) every day.**

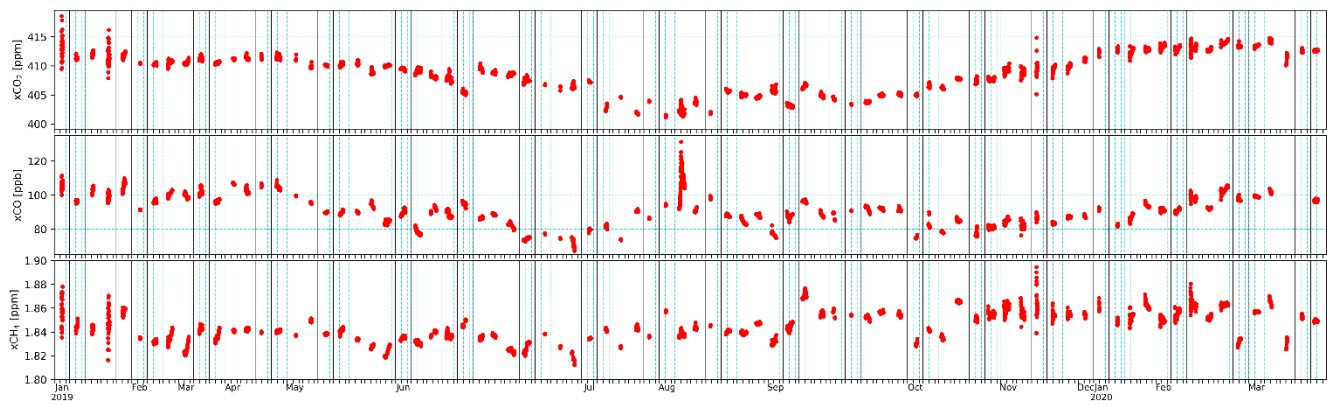

**Figure 5: Time series for $XCO_2$, XCO and $XCH_4$ obtained in Peterhof during the continuous campaign.**



### 2.3.2 Yekaterinburg (56.8°N, 60.6°E)

It was planned that immediately after the EMME campaign, the instrument FTS#84 would be transported to Yekaterinburg. Unfortunately, unforeseen organizational problems significantly delayed moving the instrument from St. Petersburg to Yekaterinburg. The instrument was finally put in operation in Yekaterinburg in October 2019 and kept measuring until the

very last day before being shipped back to KIT (April 2020). The instrument was operated at the Climate and Environmental Physics Laboratory INSMA of the Ural Federal University (UrFU). The instrument was set up in an internal yard of UrFU building. However, the building structure, which blocked the sunlight, was a limitation. Sometimes high trucks passing through the yard blocked the field of view of the instrument (See Figure 6). The spectrometer rested on the windowsill of the basement, so it was located exactly at ground level ~260 m. Under good weather conditions, measurements were carried out

approximately between 11:00 and 14:30, local time. In total, twenty-two days of measurements were collected as it can be seen in Figure 7.

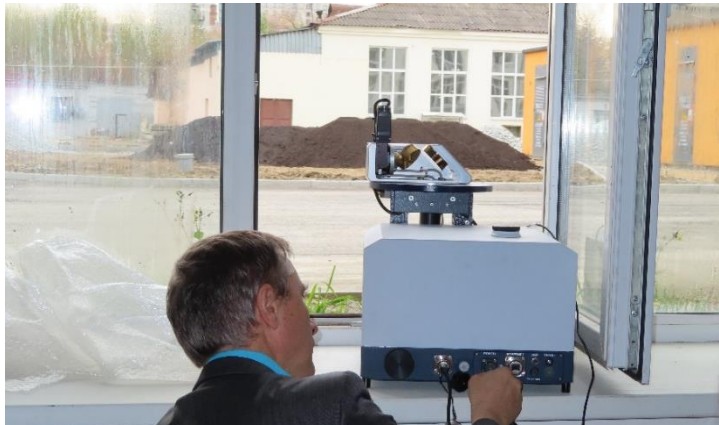

**Figure 6: Instrument setup at Yekaterinburg. The time interval of the daily measurements was constrained by the building structure, which blocked the sunlight.**

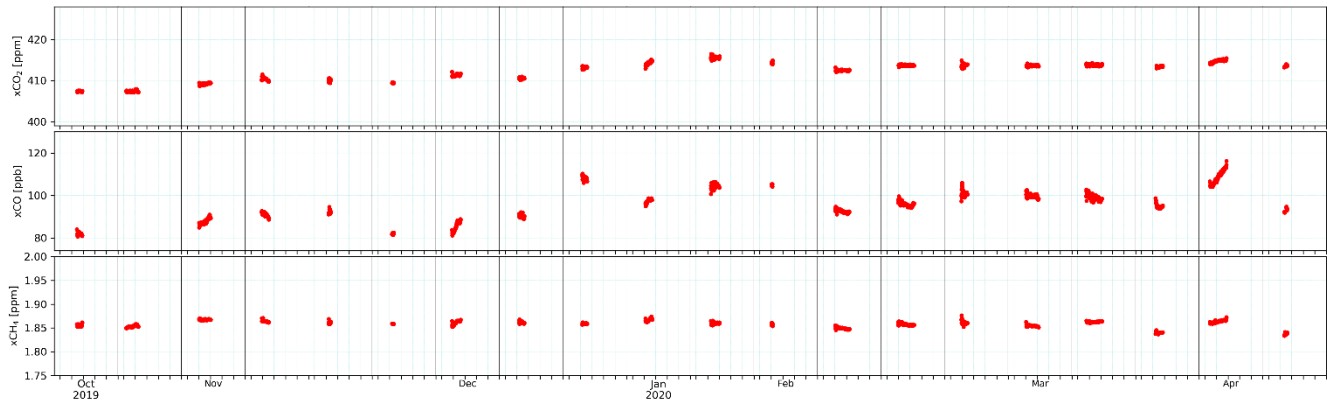


**Figure 7: Times series of XCO₂, XCO and XCH₄ observed at Yekaterinburg.**



# 3 Data sets

## 3.1 Ground-based data

### 3.1.1 COCCON

Recently, the COCCON Network (https://www.imk-asf.kit.edu/english/COCCON.php, last access: 13 May, 2021; Frey et al., 2019) was established by continuous support granted by European Space Agency (ESA). COCCON provides a supporting infrastructure for GHG measurements using the EM27/SUN spectrometer and ensures common standards for instrumental quality management and data analysis. The EM27/SUN spectrometer was developed by KIT in cooperation with the Bruker company in 2011 (Gisi et al., 2012). A second detector channel for XCO observations was added in 2015 (Hase et al., 2016).

The EM27/SUN spectrometers are widely used and there are currently about 78 instruments globally operated by different research groups. It has been shown in several studies that the results for these GHGs observed by COCCON instruments are in good agreement with official TCCON results (Frey et al., 2020; Sha et al., 2020). With the characteristics of compactness, robustness and portability, these instruments have been successfully used in several field campaigns and continuous deployments (Hase et al., 2015; Klappenbach et al., 2015; Chen et al., 2016; Butz et al., 2017; Sha et al., 2019; Vogel et al.,

2019; Tu et al., 2020, 2021a, b; Jacobs et al., 2020; Frey et al., 2021;). A preprocessing tool and the PROFFAST non-linear least squares fitting algorithm are used for data retrieval. This processing software was created in the framework of the ESA COCCON-PROCEEDS and COCCON-PROCEEDS II projects. The solar zenith angle (SZA) range of COCCON data used in this study is restricted to ≤70° in order to limit uncertainties connected to spectra recorded at very high air-mass.

## 3.2 Space-borne data

### 3.2.1 TROPOMI

The Sentinel-5 Precursor (S5-P) satellite with the Tropospheric Monitoring Instrument (TROPOMI) on board as a single payload was launched in October 2017. S5-P is a low Earth orbit polar satellite. It aims at monitoring air quality, climate and ozone layer with high spatio-temporal resolution and daily global coverage during an operational lifespan of 7 years (Veefkind et al., 2012). TROPOMI is a nadir viewing grating-based imaging spectrometer, measuring back-scattered solar radiation

spectra with an unprecedented resolution of $7 \times 7$ km$^2$ (upgraded to $5.5 \times 7$ km$^2$ in August 2019, Lorente et al., 2021). In this study, we use the improved TROPOMI XCH$_4$ product derived with the RemoTeC full-physics algorithm (Lorente et al., 2021) and apply the recommended quality value (qa) = 1.0 to the data. For CO, the SICOR (short-wave infrared CO algorithm) is deployed to retrieve the total column density of CO from TROPOMI spectra at 2.3μm (Landgraf et al., 2016; Borsdorff et al., 2018a, b). XCO is computed by dividing the CO total column by the dry air column extracted from co-located CH$_4$ file, which

reports the European Center for Medium-Range Weather Forecast (ECMWF) pressure fields. H$_2$O retrievals are also performed with SICOR algorithm. A similar quality filter is applied to the H$_2$O product as used in Schneider et al., 2020.



### 3.2.2 OCO-2

The Orbiting Carbon Observatory-2 (OCO-2) is a NASA satellite, launched in July 2014, providing space-based measurements of atmospheric $CO_2$ (Eldering et al., 2017). These observations have the potential capability to detect $CO_2$
sources and sinks with unprecedented spatial and temporal coverage and resolution (Crisp, 2015). The OCO-2 mission carries a single instrument incorporated with three high-resolution imaging grating spectrometers, collecting spectra from reflected sunlight by the surface of Earth in the molecular oxygen ($O_2$) A band at 0.764 μm and two $CO_2$ bands at 1.61 and 2.06 μm (Osterman et al., 2020). The OCO-2 satellite has three viewing modes (nadir, glint and target) and a near-repeat cycle of 16 days (98.8 min per orbit, 233 orbits in total). It samples at a local time of about 1:30 pm. The current version (V10r) of the
OCO-2 Level 2 (L2) data product, containing bias-corrected $XCO_2$ is used in this study.

In addition to the operational $XCO_2$ product derived from OCO-2 observations described above, the data product generated using the Fast atmOspheric traCe gAs retrieval (FOCAL) algorithm described in Reuter et al. (2017a, 2017b) had been used. Compared with collocated TCCON observations, the OCO-2 FOCAL data show a regional-scale bias of about 0.6 ppm and single measurement precision of 1.5 ppm (Reuter and Buchwitz, 2021). In this study, the latest version (v09) covering the time
period of 2015 – 2020 is utilized for further comparison with the COCCON results.

### 3.2.3 MUSICA IASI

The Infrared Atmospheric Sounding Interferometer (IASI) is a payload on board the EMETSAT Metop series of polar orbiting satellites (Clerbaux, 2009). The IASI instrument is a Fourier Transform Spectrometer that measures infrared radiation emitted from the Earth and emitted and absorbed by the atmosphere. It provides unprecedented accuracy and resolution on
atmospheric humidity profile, as well as total column-integrated CO, $CH_4$ and other compounds twice a day. There are currently three IASI instruments in operation on Metop-A, B and C, launched in 2006, 2012 and 2018, respectively. The MUSICA IASI retrievals are based on a nadir version of PROFFIT (Schneider and Hase, 2009), which has been developed in support of the MUSICA project. More details can be found in Schneider and Hase (2011) and Schneider et al. (2021b). A validation of the MUSICA IASI $H_2O$ profile data is presented by Borger et al. (2018).

### 250 3.2.4 GOSAT

The Greenhouse Gases Observing Satellite (GOSAT) was launched in January 2009, equipped with two instruments (the Thermal And Near-infrared Sensor for carbon Observation Fourier Transform Spectrometer (TANSO-FTS) and the TANSO Cloud and Aerosol Imager (TANSO-CAI)) (Kuze et al., 2009). The satellite is placed on a sun-synchronous orbit and passes the same point on Earth every three days. The GOSAT is the first mission to monitor the global distribution and sinks and
sources of GHGs. For this study, GOSAT FTS Short Wave InfraRed (SWIR) Level 2 data version V02.90 from National Institute of Environmental Studies (NIES) is used.



### 3.3 CAMS data

### 3.3.1 CAMS inversion

Copernicus Atmosphere Monitoring Service (CAMS) is operated by the European Centre for Medium-Range Weather
Forecasts (ECMWF), providing global inversion-optimised GHG concentration products which are updated once or twice per
year. For $XCO_2$ and $XCH_4$, the latest version data sets (v20r1 for $XCO_2$ and v19r1 for $XCH_4$) using surface air-sample as
observations input are used in this study. The CAMS global $CO_2$ atmospheric inversion product is generated by the inversion
system, called PyVAR (Python VARiational) with a horizontal resolution of $1.875^{o} \times 3.75^{o}$ and temporal resolution of 3 hours
(Chevallier, 2020a, b). The latest version (V20r1) was released in December 2020, covering the period from January 1979 to
May 2020. The V20r1 model data fits TCCON retrievals well with less than 1 ppm of absolute biases (Chevallier, 2020b).

For $XCH_4$ we used the latest version V19r1 based on inversion of surface observations only, covering the period between
January 1990 and December 2019. The CAMS $XCH_4$ inversion product are based on the TM5-4DVAR (four-dimensional
variational) inverse modelling system (Bergamaschi et al., 2010, 2013; Meirink et al., 2008) with a horizontal resolution of 2
$^{o} \times 3^{o}$ and temporal resolution of 6 hours (Segers, 2020a, b). Compared to previous releases, v19r1 data has been adjusted by
using new atmospheric $CH_4$ sinks and updated wetland emissions, and the monthly bias is usually less than 10 ppb with respect
to the TCCON network (Segers, 2020b).

### 3.3.2 CAMS reanalysis (control run)

This study aims to compare XCO retrieved from the COCCON measurements with XCO from different satellite and CAMS
data sets as well. However, no XCO is available from the before-mentioned CAMS data. Fortunately, CAMS also provides
reanalysis data sets, covering the period of 2003 – June 2020. The standard CAMS reanalysis data uses 4DVar data assimilation
in CY42R1 of ECMWF's Integrated Forecast System (IFS) (Flemming et al., 2017; Inness et al., 2019). The CAMS reanalysis
CO profiles under a control run, i.e. without any data assimilation, is obtained from Copernicus Support team. This control run
reanalysis CO profiles are using only one IFS cycle with a $0.1^{o} \times 0.1^{\circ}$ latitude/longitude resolution, 3 hours of temporal
resolution and 25 pressure levels. XCO is obtained when integrating the profiles from the lowest to the highest pressure level.

### 4 Results and discussion

### 4.1 Seasonal variability of GHGs

### 4.1.1 Peterhof

The seasonal patterns of the retrieved GHGs are shown in Figure 8, which illustrates the time series of daily mean of $XCO_2$,
$XCH_4$, XCO and $XH_2O$ from different data products at Peterhof. The CAMS-COCCON data product presented in Figure 8
and Figure 9 are discussed in section 4.3. The TROPOMI satellite has a higher spatial resolution and therefore, the available





retrieved species from TROPOMI were daily averaged within a collocation radius of 50 km around Peterhof. For the GOSAT and MUSICA IASI data sets, a collocation radius of 100 km around Peterhof is used, and for OCO-2 data, a collocation radius of 200 km is used. The measurements from the different ground- and space-based observations and model data generally show good agreements and similar seasonal variability.

COCCON $XCO_2$ is biased low by about 0.81 -3.1 ppm in comparison to CAMS and other satellite products. GOSAT (Figure 8 (a)) also shows some obvious outliers compared to the other products, which have similar behaviours. The amount of $XCO_2$ varies along the year and much of this variation is driven by respiration, which never stops but increases between fall and winter due to reduced uptake (no photosynthesis). In this case the atmospheric $XCO_2$ concentration is stable between January and April. It started to decrease from May to end of July, during which the growing season and the photosynthetic activities increase. Similar behaviour in 2019 was also observed by Timofeyev et al. (2021) and in previous years by Timofeyev et al. (2019) and Nikitenko et al. (2020). The amount of $XCO_2$ stays around 403 ppm between end of July and middle of September and starts to increase afterwards.

For $XCH_4$ COCCON shows similar a behaviour as TROPOMI and CAMS. Slightly higher mean values and variability can be seen in GOSAT $XCH_4$ with a few outliers. Compared to $XCO_2$, $XCH_4$ shows generally less seasonal variabilities with more short-term enhancements. The seasonal variation is comparable to the results of Gavrilov et al. (2014), Makarova et al. (2015a, 2015b) and Timofeyev et al. (2016). A slightly higher $XCH_4$ is observed at the end of 2019 for all data products.

XCO shows seasonal variability with a maximal value of 110 ppb in late April and decreases by nearly 40% to 70 ppb in the beginning of July. A secondary local maximal reaching ~95 ppb occurs in August. This feature needs further investigation. The COCCON XCO matches well to the CAMS reanalysis data. Moreover, the COCCON agrees better with the TROPOMI data in summer than in spring and late autumn, when TROPOMI measured higher values.

$XH_2O$ shows a strong seasonal cycle with a maximal amount of ~4700 ppm in summer and minimal amount of ~320 ppm in winter. All products show quite similar behavior with high variability, which is similar to those in Semenov et al. (2015), Timofeyev et al. (2016) and Virolainen et al. (2016, 2017). The GOSAT data have higher mean values since the measurement period covers only the time period from later spring to summer, during which higher $XH_2O$ is observed.






**Figure 8 Time series of daily mean of (a) XCO₂, (b) XCH₄, (c) XCO and (d) XH₂O for different data products at Peterhof.**



### 4.1.2 Yekaterinburg

The measurement period covered winter and spring, from 5 October 2019 to 17 April 2020 at Yekaterinburg (Figure 9). Here we use a larger radius (100 km) to collect the TROPOMI observations because there are much less overpasses at Yekaterinburg
during this period.

$XCO_2$ shows a clearly increasing tendency from October of 408 ppm to a maximal value of 415 ppm in the middle of February, which covers later autumn and winter. This is because on top of the increase due to the anthropogenic emissions there are variations due to the photosynthesis and respiration (https://atmosphere.copernicus.eu/carbon-dioxide-levels-are-rising-it-really-simple, last access: 2 July 2021). During that period the plants notably reduce or stop the photosynthesis
processes which could increase the amount of $CO_2$ in the atmosphere. Later this maximal value stays constant until mid of March. It tends to decrease and a similar behavior is observed in Peterhof.

For $XCH_4$, COCCON shows a good agreement with CAMS data, though there are not so many COCCON observations. $XCH_4$ shows generally decreasing tendency but with more short-term variabilities. Such variabilities are observed in Peterhof as well. A few TROPOMI observations in October are deviating from the other two data sets and it seems that TROPOMI
underestimates $XCH_4$. This might be because most TROPOMI measurements are located in the rim of the collecting radius and thus away from the location of Yekaterinburg, introducing some errors (see Figure A- 5). Further, this underestimation could be due to the difficulty for retrieving $CH_4$ in low- and high-albedo scenes (Lorente et al., 2021).

XCO shows in general a similar behavior of $XCO_2$, with a steady increase during late autumn and winter. It seems that the increasing behavior of XCO has an inverse relationship with $XCH_4$. This is probably due to the fact that atmospheric CO is
mainly produced by incomplete combustion of fossil fuels (Kasischke and Bruhwiler, 2002) and the oxidation of methane (Cullis et al., 1983).

As expected, most of $XH_2O$ values are below 1000 ppm, similar to Peterhof in that period. This can be explained by the saturation concentration of water vapor in air, which reduces for lower temperatures.







Figure 9 Time series of daily mean of (a) XCO₂, (b) XCH₄, (c) XCO and (d) XH₂O for different data products at Yekaterinburg.






**4.2 Correlation between COCCON and satellite products**

Figure 10 to Figure 13 show the correlations between COCCON and different satellite products at Peterhof (triangle symbols) and at Yekaterinburg (dot symbols). The satellite products and CAMS generally agrees well with COCCON and the scaling factor (slope of the fitting line; intercept is forced to 0) varies from 0.9712 to 1.0842. Figure 14 illustrates the averaged bias

and standard deviation of each product of the coincident Xgas ($XCO_2$, $XCH_4$ and $XCO$) values (in space-time) with respect to COCCON for the available gases at both sites. In order to find the coincident COCCON data, the mean value of the observations 2 hours before and after a centralized time reference is taken. Such time reference differs for each of the products as follows: the overpass time for satellite, each of the timestamp for CAMS.

The measuring period at Yekaterinburg for COCCON was mostly in winter and early spring, from October 2019 to April

2020, in which there were less sunny days. This results in less COCCON and satellite observations. There is only one coincident point between COCCON and NASA operational OCO-2 (Figure 11 (c)) and no coincident between COCCON and OCO-2 FOCAL and GOSAT products at Yekaterinburg. Even a much larger collection circle with a radius of 100 km is used for TROPOMI at Yekaterinburg, the coincidence measurements are lesser than those in Peterhof, where more than one year of measurements were performed.

Due to the short period of ground-based measurements, poor weather condition, and poorer coverage of satellites at high latitude    in    the    winter    hemisphere    (OCO-2:    Patra    et    al.,    2017    and    GOSAT: http://www.gosat.nies.go.jp/en/about_%EF%BC%92_observe.html, last access: 28 June, 2021), it becomes more difficult to validate satellite products with ground-based measurements at locations like Yekaterinburg.

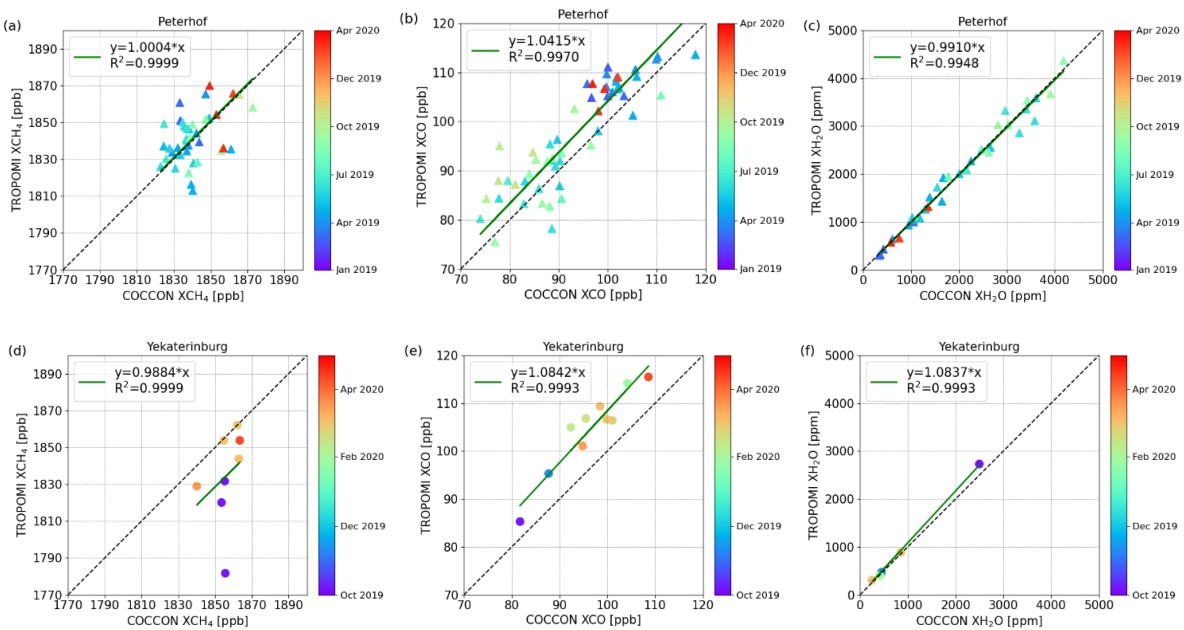

**Figure 10 Correlation plots between TROPOMI and COCCON for XCH₄, XCO and XH₂O at Peterhof (a-c) and at Yekaterinburg (d-f). The fitting lines are forced to through origin (the R² values refer to the forced fit).**



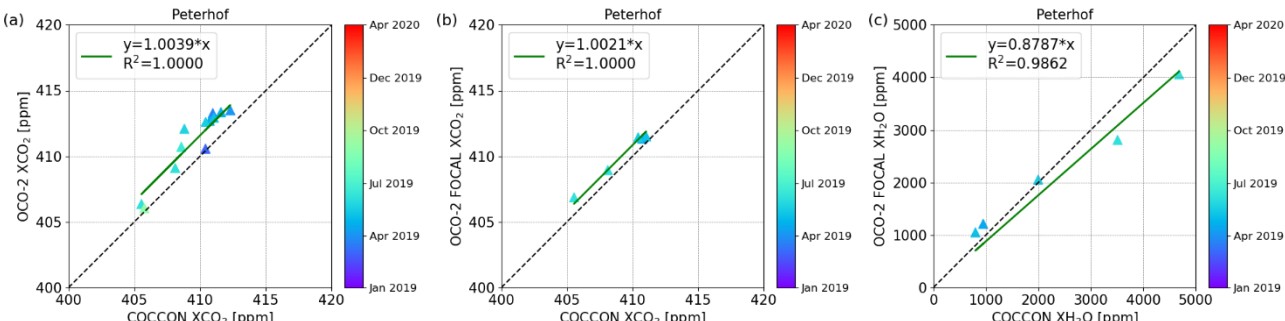

**Figure 11 Correlation plots (a-b) between NASA's operational and the FOCAL OCO-2 product and COCCON for XCO₂ and (c) between OCO-2 FOCAL and COCCON for XH₂O at Peterhof. The fitting lines are forced to through origin (the R² values refer to the forced fit).**

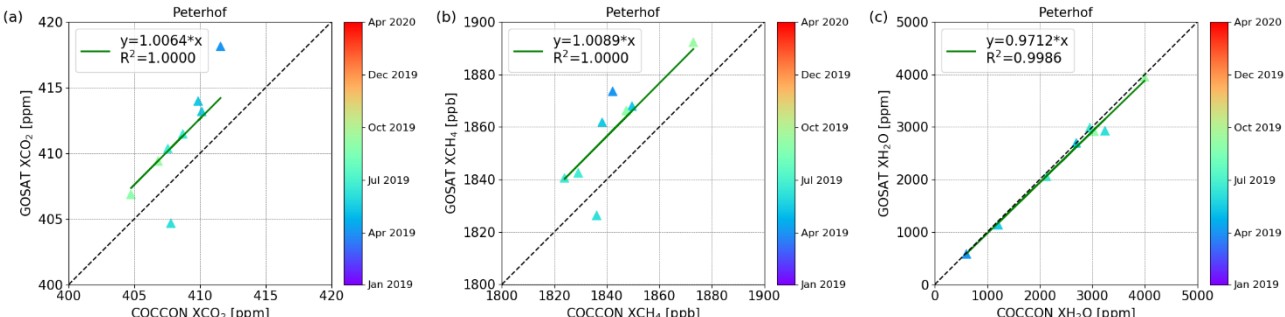

**Figure 12 Correlation plots between GOSAT and COCCON for (a) XCH₄, (b) XCO and (c) XH₂O at Peterhof. The fitting lines are forced to through origin (the R² values refer to the forced fit).**

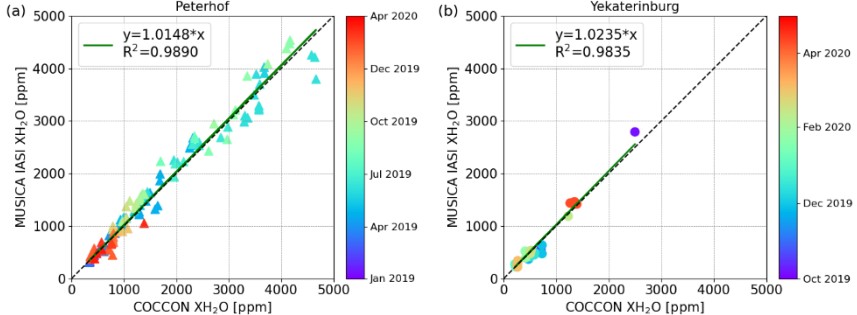

**Figure 13 Correlation plots of XH₂O between MUSICA IASI and COCCON at (a) Peterhof and (b) Yekaterinburg. The fitting lines are forced to through origin (the R² values refer to the forced fit).**

At Peterhof OCO-2 FOCAL XCO₂ data have the lowest bias with respect to COCCON, while GOSAT data show the highest bias and standard deviation (3.1 ppm ± 2.9 ppm, Figure 14). NASA operational OCO-2 and CAMS show similar biases. CAMS, TROPOMI and GOSAT measure higher XCH₄ than COCCON, among which GOSAT has the highest biases at Peterhof. The high negative bias in TROPOMI at Yekaterinburg is mainly due to the underestimation of the TROPOMI product in October, 2019. At both sites TROPOMI XCO shows higher biases than CAMS with respect to COCCON, which can be





seen in Figure 8 (c) and Figure 9 (c) – TROPOMI with higher values than COCCON. TROPOMI and GOSAT generally measure lower $XH_2O$ than COCCON, whereas MUSICA IASI shows high bias and standard deviation. However, good

correlations can be found between satellite $XH_2O$ and COCCON in Figure 10 (c) and (f), Figure 12 (c) and Figure 13.

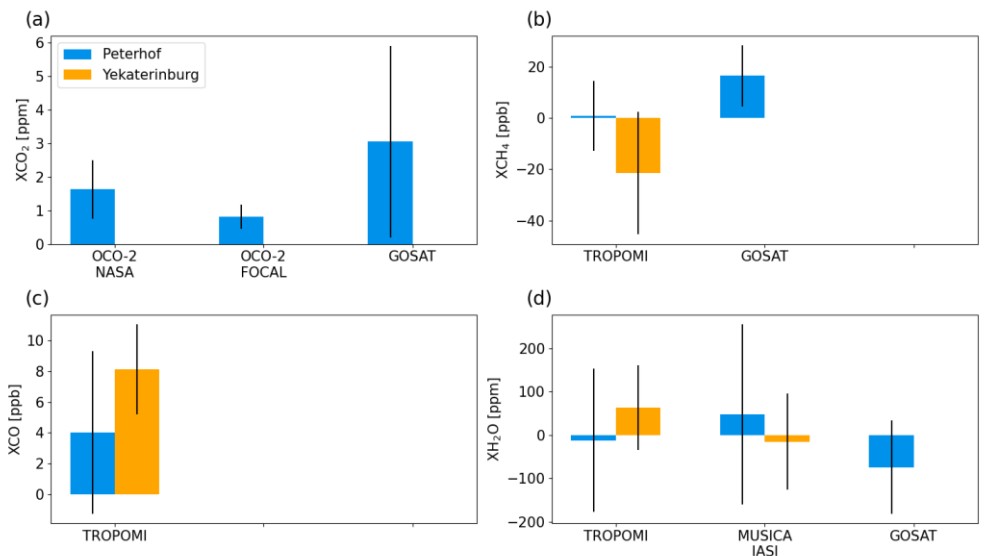

**Figure 14 Bar plots of the averaged bias derived from different products with respect to COCCON for (a) $XCO_2$, (b) $XCH_4$, (c) XCO and (d) $XH_2O$ at Peterhof and Yekaterinburg. The error bars represent the standard deviation of the averaged bias.**

### 4.3 Using CAMS model fields for upscaling COCCON observations

Unfortunately, during the continuous campaign carried out at Peterhof and Yekaterinburg, there are just a few coincident measurement days with satellite observations, especially in comparison with GOSAT and OCO-2 (see Figure 14). Although these satellites offer a global coverage, for our measurement period even with quite relaxed coincidence criteria, the comparisons do not use the majority of the ground-based observations. This is especially the case in Yekaterinburg during the observations from October 2019 to April 2020, i.e. GOSAT and OCO-2 have none or just a couple of measurements in winter

and early spring period at high latitudes. Even in Peterhof where more than one year of measurements were taken, the coincident measurements between the aforementioned satellites are rather few.

For that reason, we employ a novel method, which uses model fields for upscaling the ground-based FTIR measurements, thereby generating additional virtual coincidences. Such upscaling does not use one global scaling factor, but a time resolved one, as it is shown in Figure A- 9, Figure A- 10 and Figure A- 11 in the appendix. Although some noise is superimposed on

the temporal evolution of scaling factors, a seasonal cycle becomes apparent.

In a first step, CAMS model data are adjusted to match the value by COCCON. Then, the adjusted model fields are compared with the available satellite results data for $XCO_2$, $XCH_4$ and XCO. The assumption of this method is that the bias of the model field is a smooth function in space and time, which seems well justified due to the long atmospheric lifetime of the gases under





consideration. Since the model considers all relevant aspects of dynamics (advection, changes in tropopause altitude) and
attempts to even reproduce abundance changes due to sources and sinks, we expect that our approach is superior to ad-hoc
schemes typically used for enlarging the colocation area (as, e.g. using the potential temperature, see Keppel-Aleks et al.,
2011). In order to avoid circular reasoning in the validation based on the adjusted model fields, the method should avoid model
simulations which include the assimilation of satellite data.

### 4.3.1 Generation of the CAMS fields adjusted to COCCON observations

CAMS inversion results with surface air-sampled observations as input had been used for $XCO_2$ and $XCH_4$ (Segers, 2020a).
Unfortunately, no XCO is available on that model run. No XCO product from CAMS disable us to compare one of the main
data product of S5-P (XCO), which offers a greater number of measurements with a high horizontal resolution in comparison
with any other satellites. Instead, the CAMS team has provided special profiles of CO from CAMS reanalysis data (control
run). On that run two important points have to be mentioned: (1) no total columns for $CO_2$ and $CH_4$ were available from this
special data set and (2) no satellite data had been assimilated. Such results are available on a daily basis as described in Table
3. CAMS inversion is available on a daily basis for $XCO_2$ and $XCH_4$ but with different time frames. Unfortunately, there are
no $XCH_4$ results from CAMS for 2020, which adds a new constraint when simply comparing both results, especially for
Yekaterinburg where approximately four out of six months were measured in 2020.

**Table 3: Time range and usual daily time frame of the analysed results from CAMS and COCCON.**

| Specie | Method | Measurements availability | Time frame [UTC] |
|---|---|---|---|
| $XCO_2$ | CAMS inversion | 01-01-1979 to 31-12-2020 | 00:00 – 21:00; each 3 hours |
| | COCCON: Peterhof | 21-01-2019 to 17-03-2020 | ~ 9:00 – 13:00 |
| | COCCON: Yekaterinburg | 05-10-2019 to 17-04-2020 | ~ 6:00 – 09:00 |
| $XCH_4$ | CAMS inversion | 01-01-1990 to 31-12-2019 | 00:00 – 18:00; each 6 hours |
| | COCCON: Peterhof | 21-01-2019 to 17-03-2020 | ~ 9:00 – 13:00 |
| | COCCON: Yekaterinburg | 05-10-2019 to 17-04-2020 | ~ 6:00 – 09:00 |
| XCO | CAMS reanalysis (control run) | 01-01-2003 to 31-12-2020 | 00:00 – 21:00; each 3 hours |
| | COCCON: Peterhof | 21-01-2019 to 17-03-2020 | ~ 9:00 – 13:00 |
| | COCCON: Yekaterinburg | 05-10-2019 to 17-04-2020 | ~ 6:00 – 09:00 |

As explained before, the main idea is to adjust $XCO_2$, $XCH_4$ and XCO from CAMS by using COCCON results. This is achieved
by performing a time-resolved scaling of the model data, which is informed by the available ground-based observations. The
detailed workflow encompasses these steps:

1. As shown in Table 3, CAMS $XCO_2$ and $XCH_4$ are available on a daily basis in different prescribed time frames, while
   COCCON results are only available when specific conditions were fulfilled: good weather conditions (sunny or





almost sunny conditions), no mobile campaign and manpower available to start the measurements because the instruments were manually operated. These conditions made the measurements rather sparse but nevertheless there still is a significant number of measurements available. Therefore, the first step is to find the coincident days between CAMS and COCCON and then the COCCON results are averaged around each CAMS time if available. As the COCCON observations require sunlight, all CAMS points before 06:00 UTC and later than 18:00 UTC were filtered out. For the aforementioned each averaged CAMS time was considered as reference and all the COCCON results ± 2 hours were averaged as the coincident data. After these steps, we have both results on the same time gridding.

2.  The output from the first step are time series with coincident measurement days and time frames. These time series, which have the same date boundaries, are then divided into n smaller intervals or sub-windows. These sub-windows have the characteristics of being non-overlapping and they form equally sized bins on the time axis, as defined in the Eq. 1. The user only needs to define the number of sub-windows "n".

$$\Delta t = \frac{DT_{initial} - DT_{final}}{n}$$

Eq. 1

3.  Additionally, a sliding-sub-window, with the same size described in step 2, is run over both time series with the main difference of being shifted by half of the size of the initial sub-window but still being not overlapping between them. Therefore, after step 2, the step 3 is done in order to look at the neighbours.

4.  In each of these sub-windows (described above: step 2 and 3) a correlation analysis is carried out independently of the other sub-windows. In order to make the COCCON time series adjust better to CAMS results, a linear correlation with the intercept forced to zero is carried out and therefore the slope gives the scaling factor for the CAMS data.

5.  Each sub-window defined in step 2 is taken as a base with its slope calculated in step 4. After that, the slopes in the neighbourhood are also calculated in each overlapping sub-window defined in step 3, Finally, all the slopes are then averaged. Such averaged slope represents the scaling factor in that sub-window. It is important to mention that this number of sub-windows (and then its size) was adjusted until good results were achieved as described below.

6.  Finally, with the scaling factor calculated in step 5, the original CAMS fields keeping their original temporal sampling are scaled in the whole range of each sub-window.

**4.3.2 Selection criteria for the best number of windows**

In order to choose the best number of windows, the scaling code is run starting from windows=1 and stops when two different conditions are fulfilled:

1.  The Root-Mean-Square-Deviation (RMSD), which is calculated with the Eq. 2 between COCCON and the CAMS-COCCON data, must be the lowest possible.

$$RMSD = \sqrt{\frac{\sum_1^t (CAMS_{Scaled} - COCCON)^2}{t}}$$

Eq. 2

2.  The number of measurements points in each of the windows must be larger than four.





The second condition is very important because if the number of windows increase, the windows size (number of measurement points) decreases until no more points are available in some windows because the distribution of measurements points in the time domain is non-homogeneous.

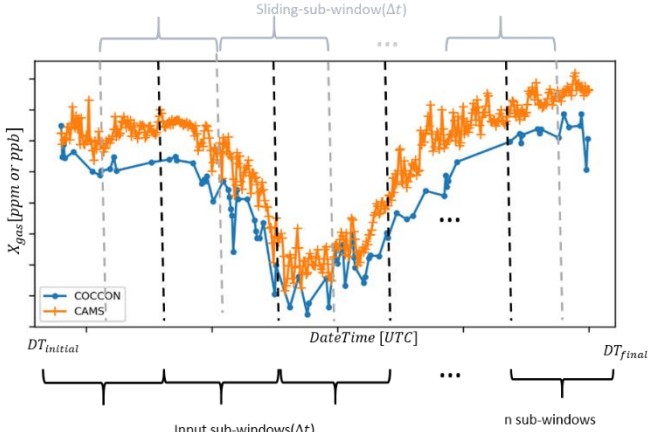

**Figure 15: Principle of the scaling method. Sub-windows are separated with black dotted line and sliding-sub-windows with grey**
**dotted line. The windows size (Δt) is defined in Eq.1.**

### 4.3.3 Verification of the method

In order to test the method before it is applied to the study area, a much denser dataset in the COCCON network is used to proof its performance. Two years of measurements (January 2018 – December 2020) taken in Karlsruhe with the instrument FTS#37 which is the reference in COCCON were selected for this purpose. For the sensitivity study, three different sub-sets
were generated from the original dataset. Such sub-sets consist of a percentage (40%, 60% and 80%) of the total amount of measurement days, which are randomly selected. This is done in order to simulate the reduced number of observations available in the study area. The GHG used for this short sensitivity study is $XCH_4$ because a comparison between each of the scaling results (for each dataset) can be compared with TROPOMI as well. The main results of this verification exercise are presented in the Figure A- 6 to Figure A- 8 in the appendix. In Figure A- 6 a plot showing the RMSD as function of the number of
windows is presented for each subset. Such results are used in order to decide the best number of windows. The correlations between CAMS and the original COCCON $XCH_4$ measurements are presented in Figure A- 7 (a), whereas Figure A- 7 (b), (c) and (d) shows the results between COCCON $XCH_4$ and its CAMS-COCCON for 40%, 60% and 80% of the original COCCON data, respectively. The satellite comparisons of the original COCCON $XCH_4$ with TROPOMI are shown in Figure A- 8 (a), whilst Figure A- 8 (b), (c) and (d) show the TROPOMI $XCH_4$ comparison but for CAMS-COCCON by using 40%, 60% and
80% of the original COCCON measurement days. The most important conclusion can be drawn from Figure A- 7 and Figure A- 9. Figure A- 7 indicates a small bias between CAMS and COCCON (of about 0.12%), which is successfully removed in the CAMS-COCCON fields, so the latter approximate the missing observational value in an optimal sense. Figure A- 9 shows the scaling factor as function of time, clarifying that the correction is not just the trivial removal of a constant bias factor, but





that some seasonal variation in the model – observation difference can be corrected as well. Note that we do not require in our
approach that the COCCON values are superior over the CAMS values. This test is performed to clarify that the CAMS fields
adjusted in the manner we described before provide the best prediction for what COCCON would have observed on a certain
date.

**4.4 Combined data results by using the scaling method**

The scaling method described above is applied to $XCO_2$, $XCH_4$ and XCO at Peterhof and Yekaterinburg. The number of
selected windows for $XCO_2$, $XCH_4$ and XCO was 11, 10, 11 at Peterhof and 5, 2, 4 at Yekaterinburg, respectively. These
scaled results are then compared with all the available satellite products as described in this study.

**4.4.1 Peterhof**

After using the scaling method, the COCCON-adjusted CAMS data show close agreement with COCCON for $XCO_2$, $XCH_4$
and XCO (see Figure A- 3).
480       The CAMS-COCCON data fill the gap during the measurements, providing a continuous period of a new intermediate or
combined (CAMS-COCCON) data product, which helps to have more coincident data with satellites observations. Figure 16
to Figure 18 show the CAMS-COCCON data in comparison to the available observations from different satellite products.
There are more coincident data points for the operational OCO-2 product than OCO-2 FOCAL $XCO_2$, which could be because
the           OCO-2           product           has           approximately           three           times           more           soundings
(https://climate.esa.int/sites/default/files/ATBDv1_OCO2_FOCAL.pdf, last access 2 July 2021). However, their correlations
and patterns are quite similar, whereas OCO-2 FOCAL shows better agreement with CAMS-COCCON data. GOSAT $XCO_2$
has a similar correlation with CAMS-COCCON as found for OCO-2 data but with some outliers. For $XCH_4$, the CAMS-
COCCON are mostly higher than TROPOMI but lower than GOSAT. The CAMS-COCCON XCO agrees well with
TROPOMI data with a $R^2$ of 0.9968.



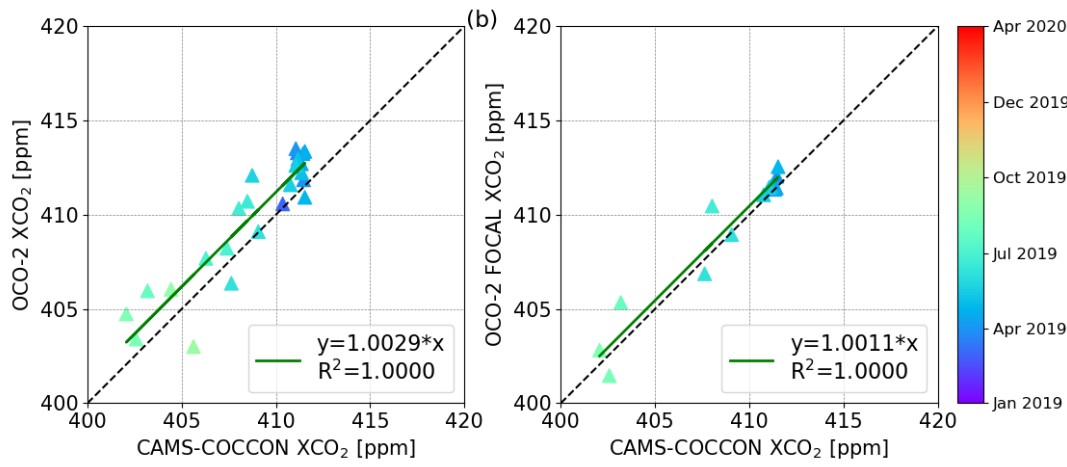


**Figure 16 Correlation plots of (a) OCO-2 and (b) OCO-2 FOCAL with respect to CAMS-COCCON $XCO_2$ at Peterhof. The fitting lines are forced to through origin (the $R^2$ values refer to the forced fit).**

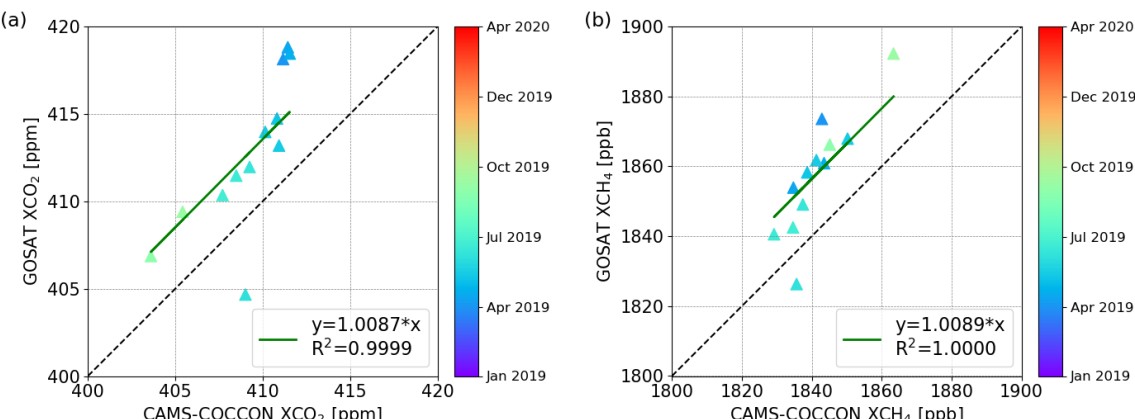

**Figure 17 Correlation plots of (a) GOSAT $XCO_2$ and (b) GOSAT $XCH_4$ with respect to CAMS-COCCON at Peterhof. The fitting lines are forced to through origin (the $R^2$ values refer to the forced fit).**






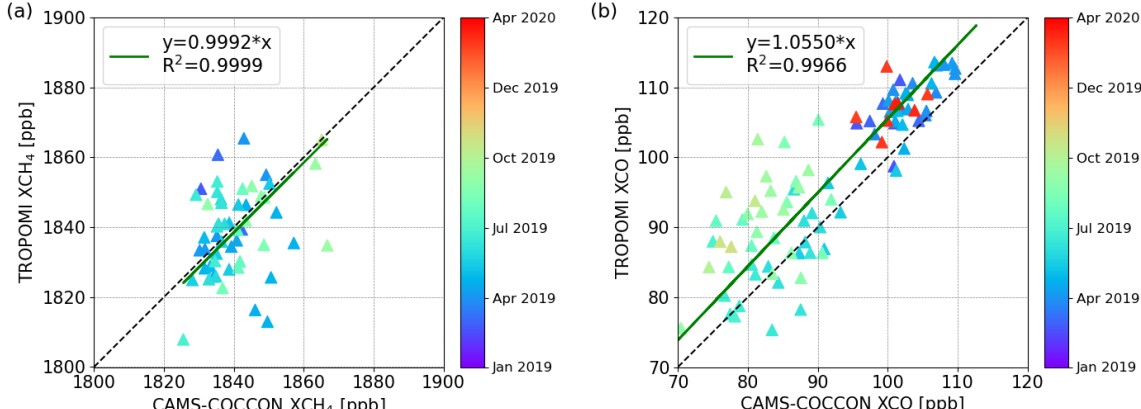

**Figure 18 Correlation plots of (a) TROPOMI XCH₄ and (b) TROPOMI XCO with respect to CAMS-COCCON at Peterhof. The fitting lines are forced to through origin (the R² values refer to the forced fit).**

### 4.4.2 Yekaterinburg

The scaled data are much more important in Yekaterinburg because in this city there are just a few coincident measurement days between COCCON spectrometer and satellite results, mainly because of the season of the measurements taken in winter and spring. That makes a real challenge in finding the best number of sub-windows to better adjust COCCON to CAMS results, which is rather small (between 2 and 3). Nevertheless, as it can be seen in Figure A- 4, the CAMS-COCCON data agree better with the coincident COCCON observations, which indicates that the scaling improves the compatibility of CAMS data with

COCCON, although the amount of sampling points is extremely small.

The correlations between the CAMS-COCCON and the OCO-2 and TROPOMI data are presented in Figure 19. There are not too many coincident data points than those at Peterhof due to the lesser COCCON and satellite observations and mostly poor weather condition in winter. The COCCON measurement ended in 17 April 2020. Here we use a larger radius (100 km) to collect TROPOMI data for coincident COCCON observations.

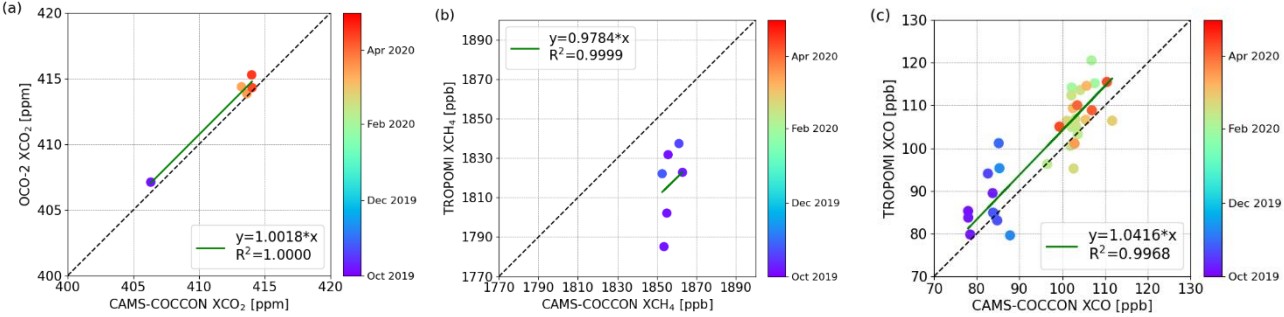


**Figure 19 Correlation plots of (a) XCO₂ between OCO-2 and CAMS-COCCON, (b) XCH₄ between TROPOMI and CAMS-COCCON, and (c) XCO between TROPOMI and CAMS-COCCON observations at Yekaterinburg. The fitting lines are forced to through origin (the R² values refer to the forced fit).**



The averaged biases between satellite products with respect to CAMS-COCCON are presented in Figure 20. Table 4

summarized selected biases and standard deviation of satellite products compared to COCCON and CAMS-COCCON data. Here, only when the coincident data between satellite observations and COCCON and CAMS-COCCON are both available (at least at one site), are shown. For $XCO_2$ the biases decrease slightly when OCO-2 is compared with COCCON and to CAMS-COCCON. The absolute bias between TROPOMI $XCH_4$ and CAMS-COCCON increased mostly twice at both sites in comparison to the direct TROPOMI $XCH_4$ to COCCON comparison. The increased low bias at Peterhof is mainly driven by

the TROPOMI outliers in April (Figure 8 (b)). The increased low bias at Yekaterinburg is due to the fact that the CAMS-COCCON data are only available up to end of 2019 and all TROPOMI data in autumn 2019 are biased low (Figure 9 (b)). For XCO the bias increased slightly at Peterhof and decreased by nearly half at Yekaterinburg when using CAMS-COCCON as the reference instead of COCCON at both sites.

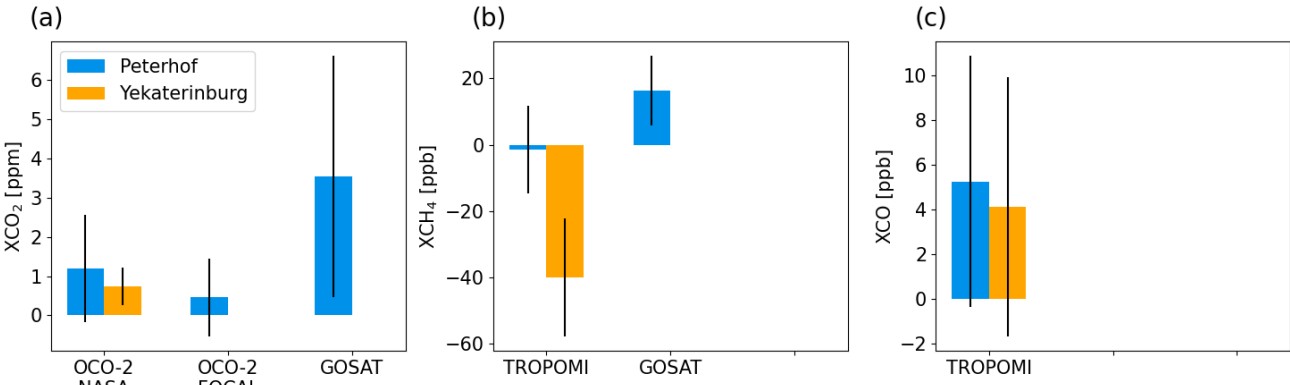

**Figure 20 Bar plots of the averaged bias derived from different products with respect to CAMS-COCCON for (a) $XCO_2$, (b) $XCH_4$ and (c) XCO at Peterhof and Yekaterinburg. The error bars represent the standard deviation of the bias.**

**Table 4 Selected averaged bias and standard deviation between satellite products and COCCON, and between satellite products and CAMS-COCCON at Peterhof and Yekaterinburg. The number of coincident results is shown in the parenthesis.**

|  |  | OCO-2 $XCO_2$ (ppm) | TROPOMI $XCH_4$ (ppb) | TROPOMI XCO (ppb) |
|---|---|---|---|---|
| Peterhof | COCCON | 1.63 ± 0.87 (15) | 0.85 ± 13.67 (39) | 4.02 ± 5.30 (54) |
|  | CAMS-COCCON | 1.19 ± 1.37 (24) | -1.49 ± 13.18 (53) | 5.26 ± 5.61 (93) |
| Yekaterinburg | COCCON | -- (1) | -21.5 ± 23.95 (8) | 8.12 ± 2.92 (11) |
|  | CAMS-COCCON | 0.74 ± 0.49 (5) | -40.04 ± 17.81 (6)* | 4.12 ± 5.80 (33) |

* No CAMS $XCH_4$ in 2020.

### 4.4.3 Gradients between Peterhof and Yekaterinburg

The gradients (ΔXgas) are the difference of each products between two sites during the same time period. The gradients between Peterhof and Yekaterinburg (Peterhof-Yekaterinburg) are presented in Figure 21. The measuring time of COCCON





at Yekaterinburg is less than that at Peterhof. We therefore use monthly means at each site to compute the gradients. A collecting circle with a radius of 100 km is used for TROPOMI at both sites. The coincident measurement days at both sites
start from October 2019 until April 2020.

For XCO$_2$ the gradients between COCCON at both sites are mostly negative and lower than those of CAMS and CAMS-COCCON data sets. Higher absolute gradients are observed in early of the year for COCCON. In November and December both CAMS and CAMS-COCCON $\Delta$XCO$_2$ show positive values whereas COCCON has negative values. This discrepancy might be due to the sparseness of COCCON observation during winter. The gradients of different data sets generally fit well
for XCH$_4$, except that of TROPOMI in October due to the low number of observations in winter. COCCON $\Delta$XCO shows highest absolute value in January, when CAMS value is near to zero. The large variations in $\Delta$XCO are in reasonable agreement with the COCCON gradients.

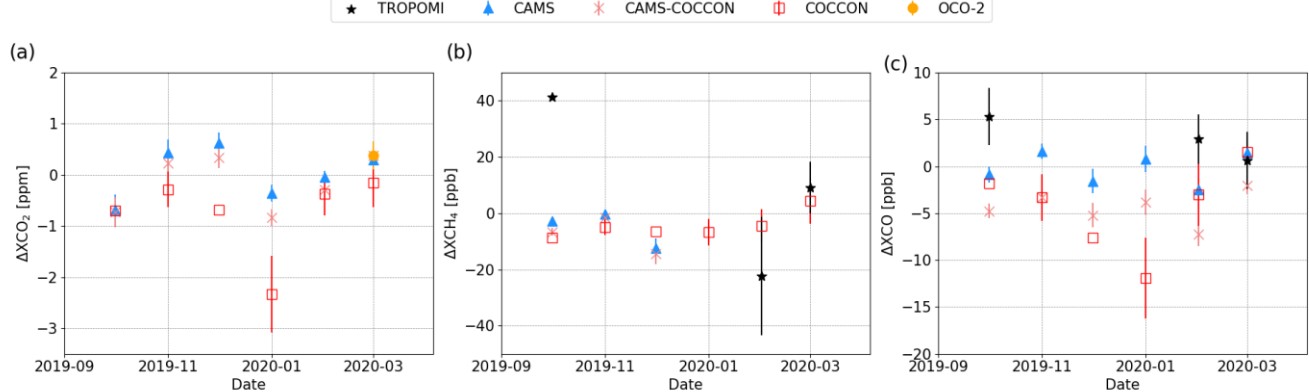

**Figure 21 Monthly mean of gradients for different gases ($\Delta$Xgas) between Peterhof and Yekaterinburg (Peterhof-Yekaterinburg)**
**for different products. The error bars are calculated based on the standard deviation at two sites.**

**4.5 St. Petersburg city emission transport event tracked by TROPOMI**

The results of the EMME campaign are in detail described and analysed in Makarova et al., (2020) and Ionov et al. (2021), nevertheless none of these studies performed any satellite comparison so far. Therefore, in this sub-section we show how a satellite with a high temporal and spatial resolution can measure and track a large transport of pollutants in a megacity like St.
Petersburg. During EMME campaign, we have been lucky to have the overpassing of the TROPOMI satellite during one of the days with strong transport gradient as presented in Makarova et al. (2020). Such results are presented in Figure 22, which illustrates the XCH$_4$ and XCO observations on a sample day on April 25, 2019 when the wind flowed from northeast to east before noon. The coincident TROPOMI data are the mean value collected within a circle of 15 km radius. The downwind COCCON instrument FTS#84 measured significant enhancements of XCH$_4$ and XCO around 9:00 UTC. The higher XCH$_4$
measured by FTS#84 than that by FTS#80 is later observed by TROPOMI as well at 10:40 UTC, though the absolute values are lower in TROPOMI than the corresponding COCCON observations. When comparing observations at two locations, the difference between them at 10:40 UTC is about 10.6 ppb measured by COCCON and 9.4 ppb by TROPOMI (Figure 22 – (e)).





For XCO, TROPOMI observes higher values than COCCON. The difference between two locations at 10:40 UTC is 9.5 ppb for COCCON and 12.5 ppb for TROPOMI. The increase of XCO at FTS#80 location measured by COCCON can also be
detected by TROPOMI, as it increased from 107.0 ppb to 115.7 ppb.

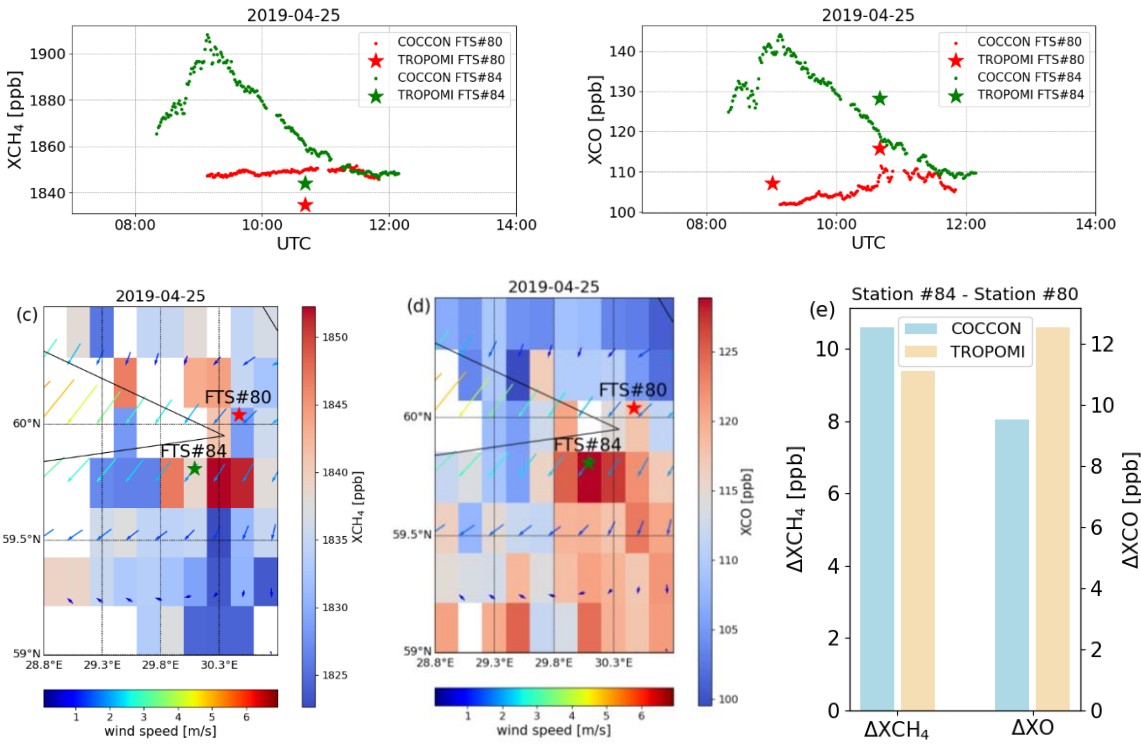

**Figure 22 Time series of COCCON and coincident TROPOMI observations for XCH$_4$ (a) and XCO (b), spatial distribution of XCH$_4$ (c) and XCO (d) on a 0.1° × 0.1° latitude/longitude grid together with the ERA5 wind at 12:00 UTC, and (e) bar plot for XCH$_4$ and
XCO gradients of COCCON and TROPOMI on April 25, 2019.**

**Conclusion**

The present study analyses ground-based COCCON and space-based TROPOMI, OCO-2, OCO-2 FOCAL, GOSAT and MUSICA IASI observations (XCO$_2$, XCH$_4$, XCO, XH$_2$O), supported by CAMS model data (XCO$_2$, XCH$_4$, XCO) in Peterhof and Yekaterinburg cities located at high latitude. Such stationary observations were performed during 2019-2020 and a mobile
city campaign was carried out in St. Petersburg in 2019 within the framework of the VERIFY project.

All the data products in Peterhof show similar seasonal variability. However, for XCO$_2$, the COCCON data set is generally lower than the other available data sets among which GOSAT has a highest standard deviation than the other datasets. TROPOMI observes slightly lower XCH$_4$ but slightly higher XCO than the other products. The largest seasonal variability is seen in XH$_2$O. Higher amounts of XH$_2$O are observed in summer mostly due to higher evaporation and precipitation, which is
expected. The averaged GOSAT XH$_2$O is higher than the other products due to its short measurement period, which is mostly



in summer. There is shorter measurement period in Yekaterinburg, covering mostly winter and spring, from October 2019 to April 2020. Similar seasonality and concentrations are observed to that in Peterhof at the same time period.

The satellite observations are sparser in the high latitude regions than in mid and low latitude regions, while models provide continuous data sets. The ground-based COCCON observations have been proved to be highly accurate by many previous studies. To combine the advantages of CAMS and COCCON data sets, we developed an upscaling method by adjusting CAMS data to the COCCON observations collected at Peterhof and Yekaterinburg to obtain a continuous data of virtual COCCON observations (as demonstrated using different sub-sets of COCCON measurements at Karlsruhe). This method is more important for Yekaterinburg, where we face three different problems: 1. less amounts of measurements in general (around 6 months compared to 15 months in Peterhof), 2. less measurement days per month (mostly in winter), and 3. shorter daily period

of measurements. As expected, the CAMS-COCCON data show better correlations with COCCON observations than the original CAMS data sets. The CAMS-COCCON data are then compared with satellite products, showing good agreements as well and generally similar biases to that between satellite products and COCCON observations. This method was also used for the observations at Yekaterinburg where less COCCON measurements were taken. The gradients between the two study sites ($\Delta$Xgas) are similar between CAMS and CAMS-COCCON data sets. There are a few COCCON and satellite $\Delta$Xgas

measurements, fitting well to that of CAMS-COCCON. These results presented in this study indicate that our scaling method is working reliably.

In addition, the $XCH_4$ and XCO observations recorded during one of the mobile city campaign days (April 25, 2019) was analyzed. In the city campaign, two COCCON instruments were set up in the upwind and downwind sites and the wind flowed from northeast to east before noon on the sample day. The downwind COCCON instrument measured obvious enhancements

in both $XCH_4$ (10.6 ppb) and XCO (9.5 ppb), which is also observed by TROPOMI (9.4 ppb in $XCH_4$ and 12.5 ppb XCO, respectively).

*Author contributions.* CA and QT developed the research question, performed the data analysis and wrote the manuscript with

support from FH. MS, BE, CD and FK provided the MUSICA data. MF supported the instrument calibration at KIT before the campaign. MVM, SCF and SIO carried out stationary observations at the Peterhof site (SPbU) and processed raw EM27/SUN data. MB and MR revised and contributed to the improvement of the manuscript. MVM, KG, VZ, DVI, FK, MMF, MS, M, MB, TB, TW proofread the manuscript.

*Data availability:* The data are accessible by contacting the authors (carlos.alberti@kit.edu and qiansi.tu@kit.edu). The OCO-2 data product is publicly available through the NASA Goddard Earth Science Data and Information Services Center (GES DISC) for distribution and archiving (http://disc.sci.gsfc.nasa.gov/OCO-2; last access: 06 May 2021). The OCO-2 FOCAL XCO2 v09 product can be obtained from the OCO-2 FOCAL website (http://www.iup.uni-bremen.de/~mreuter/focal.php; last access: 03 August 2021). The SRON S5P-RemoTeC scientific TROPOMI CH4 dataset from this study is available for



download at https://doi.org/10.5281/zenodo.4447228 (Lorente et al., 2021, last access: 06 May 2021). The S5-P H2O dataset from this study is available for download at http://ftp.sron.nl/open-access-data-2/TROPOMI/tropomi/hdo/10_3/ (Schneider et al., 2021a, last access: 06 May 2021). The S5-P NO2 and CO datasets are publicly available from https://scihub.copernicus.eu/ (last access: 06 May 2021; ESA, 2020). The access and use of any Copernicus Sentinel data available through the Copernicus Open Access Hub are governed by the legal notice on the use of Copernicus Sentinel Data and Service Information, which is

given here: https://sentinels.copernicus.eu/documents/247904/690755/Sentinel_Data_Legal_Notice (last access: 06 May 2021; European Commission, 2020). The GOSAT TANSO-FTS SWIR L2 data are available from the GOSAT Data Archive Service (GDAS) at https://data2.gosat.nies.go.jp/ (GDAS, last access: 07 July 2021).

*Competing interests.* The authors declare that they have no conflict of interest.


*Acknowledgement:* This study was supported by VERIFY project, funded by the European Union's Horizon 2020 research and innovation programme under grant agreement no. 776810. The University of Bremen received co-funding from ESA (GHG-CCI+ project) for the generation and data analysis of the FOCAL product. The CAMS results were generated using Copernicus Atmosphere Monitoring Service (2017–2020) information. Neither the European Commission nor ECMWF is

responsible for any use that may be made of the Copernicus information or data it contains. We also thank Michela Giusti in the Data Support Team at ECMWF for retrieving and providing comments about the CAMS data. This work has benefit from the project MUSICA (funded by the European Research Council under the European Community's Seventh Framework Programme (FP7/2007-2013)/ERC Grant Agreement number 256961) and from financial support in the context of the projects MOTIV and TEDDY (funded by the Deutsche Forschungsgemeinschaft under project IDs/Geschäftszeichen

290612604/GZ:SCHN1126/2-1 and 416767181/GZ:SCHN1126/5-1, respectively). MUSICA IASI retrieval calculations for this work were performed on the supercomputers ForHLR I+II and HoreKa funded by the Ministry of Science, Research and the Arts Baden-Württemberg and by the German Federal Ministry of Education and Research. K.Gribanov and V.Zakharov activity was partially supported in frame of the Russian Science Foundation, grant No. 18-11-00024-П. We are grateful to Prof. Yuri Timofeyev for his insightful comments on the manuscript.


*Financial support.* This research has been supported by the European Commission, H2020 Observation-based system for monitoring and verification of greenhouse gases (VERIFY, grant no. 776810.

The article processing charges for this open access publication were covered by a Research Centre of the Helmholtz

Association.



# Appendix

**Tabel A- 1. Overview of the satellite and model data products used in this study.**

| Data product | Species | Algorithm/ model | Product version/Level | qa | References | Data provider and data access information |
|---|---|---|---|---|---|---|
| COCCON | XCH$_4$, XCO, XH$_2$O | PROFFAST | | | Frey et al., 2019 | |
| TROPOMI | XCH$_4$ | RemoTeC | Level 2 | qa=1.0 | Lorente et al., 2021 | http://ftp.sron.nl/open-access-data-2/TROPOMI/tropomi/ch4/14_14_Lorente_et_al_2020_AMTD/ (last access: 3 May 2021) |
| | XCO | SICOR (Shortwave Infrared CO Retrieval) | offline, Level 2, v1.2 | qa=1.0 | Landgraf et al., 2016 Borsdorff et al., 2018 | https://s5phub.copernicus.eu/dhus/#/home (last access: 3 May 2021) |
| | XH$_2$O | SICOR | Level 2, v8.1 | | Schneider et al., 2021a Scheepmaker et al., 2016 | http://ftp.sron.nl/open-access-data-2/TROPOMI/tropomi/hdo/10_3/ (last access: 3 May 2021) |
| OCO-2 | XCO$_2$ | ACOS | v10r | qa = 0 | Kiel et al. 2019 Osterman et al., 2020 | Product OCO2_L2_Lite_FP 10r Obtained from NASA's Earthdata GES DISC website: https://disc.gsfc.nasa.gov/datasets/OCO2_L2_Lite_FP_10r/summary?keywords=OCO-2%20v10r (last access: 3 May 2021) |
| OCO-2 FOCAL | XCO$_2$ | FOCAL | v09 | | Reuter et al., 2017a Reuter et al., 2017b Reuter et al., 2020 | University of Bremen |
| GOSAT | XCH$_4$, XCO, XH$_2$O | | V02.90 | | Kuze et al., 2009 | https://data2.gosat.nies.go.jp/ (last access: 7 July 2021) |
| MUSICA IASI | XH$_2$O | PROFFIT (nadir version) | v3.2.1 and v3.3.0 | spectral fit quality check | Schneider et al., 2021b | https://www.imk-asf.kit.edu/english/musica-data.php |



| | | | | | according to Schneider et al., 2021 | | (last access: 7 July 2021) |
|---|---|---|---|---|---|---|---|
| CAMS | $XCO_2$ | PyVAR | v20r1 | | Chevallier, 2020a, b | https://ads.atmosphere.copernicus.eu/cdsapp#!/dataset/cams-global-greenhouse-gas-inversion?tab=form (last access: 3 May 2021) |
| | $XCH_4$ | TM5-4DVAR | v19r1 | | Segers, 2020a, b | |
| | XCO | Integrated Forecast System | control run | | Flemming et al., 2017 Inness et al., 2019 | on request |





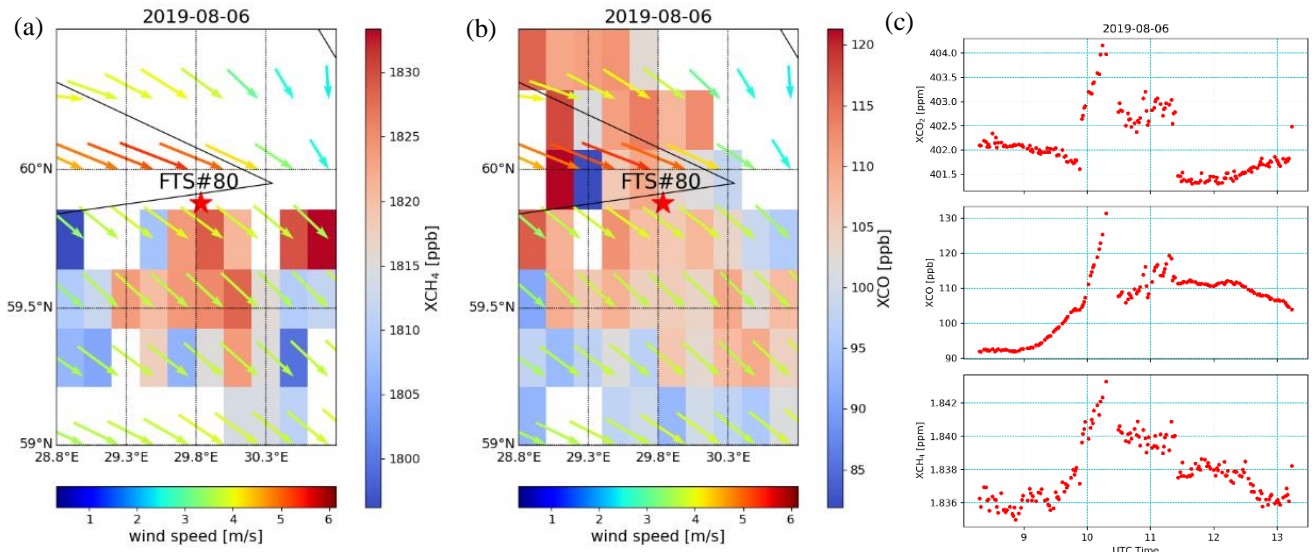

**Figure A- 1 Spatial distribution of XCH$_4$ (a) and XCO (b) on a 0.1° × 0.1° latitude/longitude grid together with the ERA5 wind at 12:00 UTC, and (c) daily time series of XCO$_2$, XCO and XCH$_4$ (from bottom to down) on 06 August, 2019.**

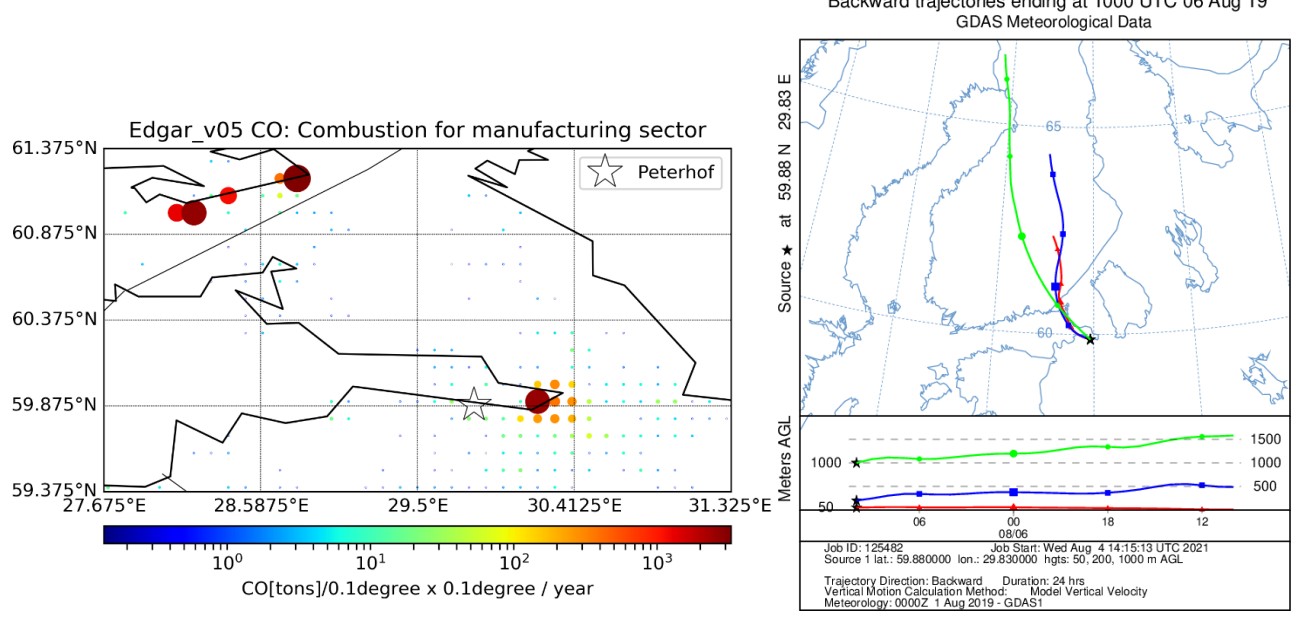

**Figure A- 2 (a) spatial distribution of CO emissions (tons/0.1 degree × 0.1 degree /year) from Sector-Specific Gridmaps: Combustion for manufacturing. Data source: EDGAR v5.0, 2015 (https://edgar.jrc.ec.europa.eu/dataset_ap50, last access: 04 August 2021) and (b) backward trajectories arriving in Peterhof on 06 August 2019, calculated by using HYSPLIT model.**



**Figure A- 3 Correlation plots of CAMS (left column) and CAMS-COCCON (right column) with respect to COCCON for XCO$_2$ (a-b), XCH$_4$ (c-d) and XCO (e-f) at Peterhof. The fitting lines are forced to through origin.**







**Figure A- 4 Correlation plots of CAMS (left column) and CAMS-COCCON (right column) with respect to COCCON for XCO$_2$ (a-b), XCH$_4$ (c-d) and XCO (e-f) at Yekaterinburg. The fitting lines are forced to through origin.**



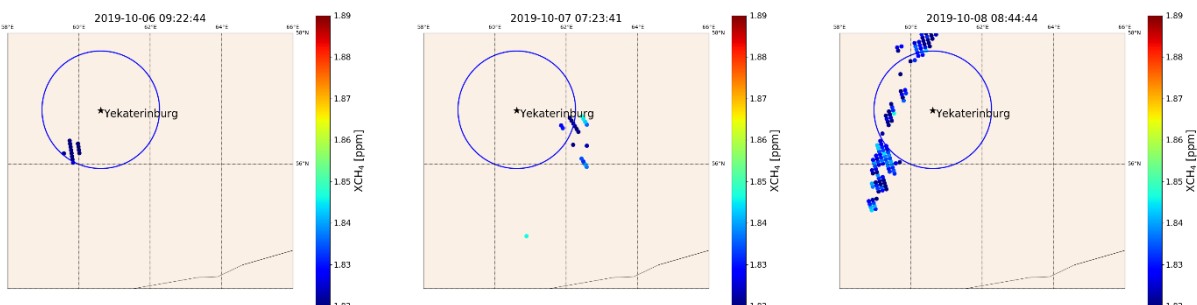

**Figure A- 5 Sample days for TROPOMI measurements (qa = 1.0) in October 2019. The circle has a radius of 100 km, centred at Yekaterinburg. The colour represents the value of XCH₄.**

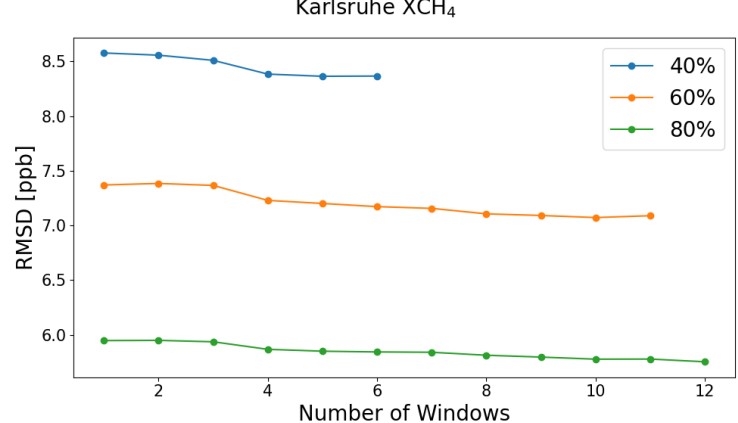


**Figure A- 6 Root-mean-square deviation between CAMS-COCCON and COCCON with respect to number of windows for XCH₄ according to 40%, 60% and 80% COCCON data points at Karlsruhe.**

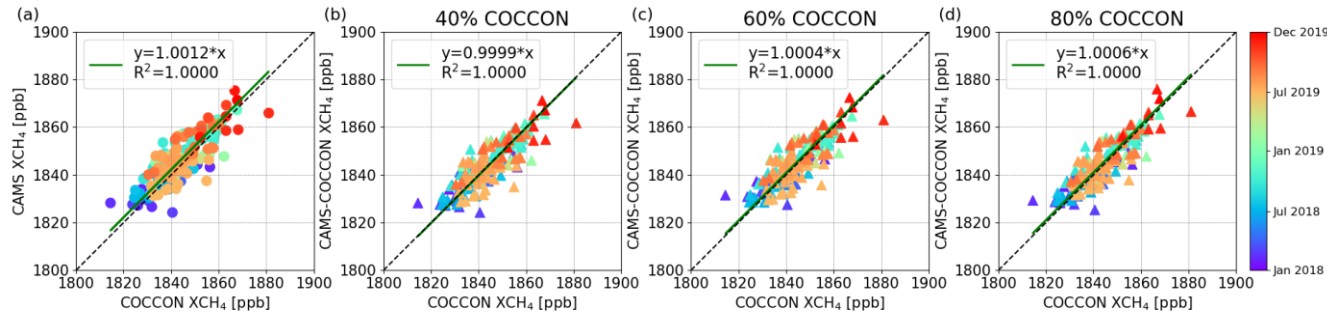

**Figure A- 7 Correlation plots of (a) CAMS and (b-d) CAMS-COCCON with respect to COCCON XCH₄ at Karlsruhe. The CAMS-**
**COCCON data sets are based on 40%, 60% and 80% of COCCON measurement days.**





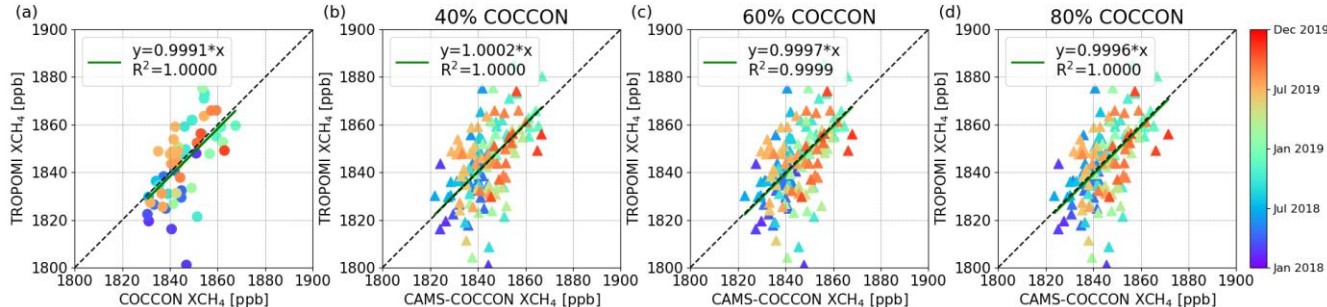

**Figure A- 8 Correlation plots of (a) COCCON and (b-d) CAMS-COCCON with respect to TROPOMI XCH₄ at Karlsruhe. The CAMS-COCCON data sets are based on 40%, 60% and 80% of COCCON measurement days.**

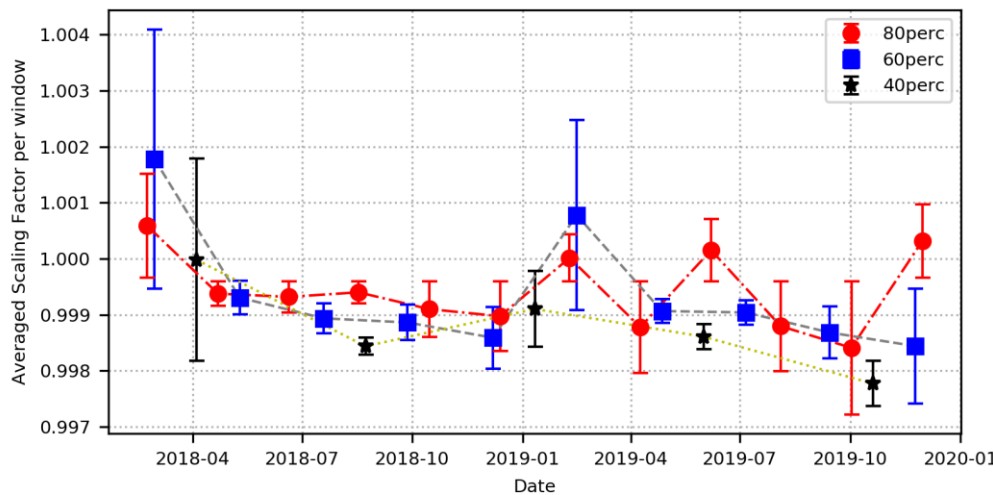


**Figure A- 9 Temporal variation of the averaged scaling factors in each sub-window for the number of windows selected for each sub-set of COCCON measurements at Karlsruhe (40%, 60% and 80% of the total measurement days with the FTS#37). The error bar represents the standard deviation calculated in each sub-window.**





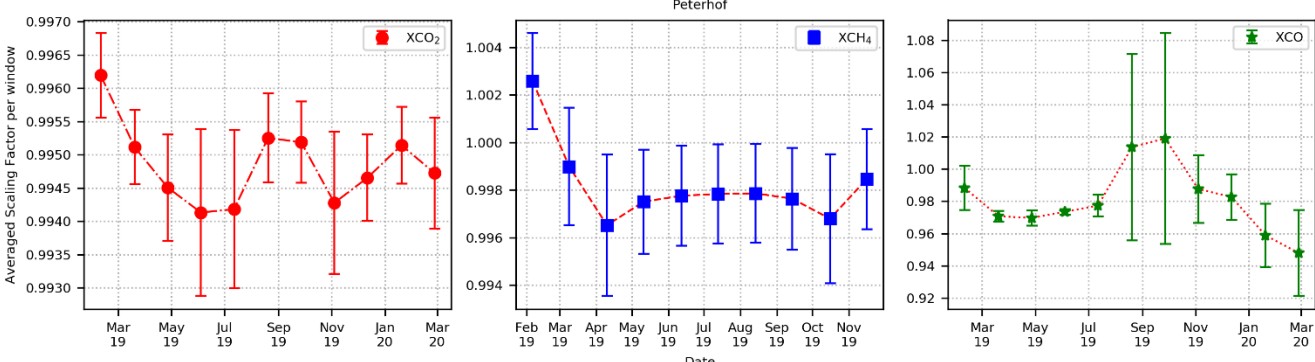

**Figure A- 10 Temporal variation of the averaged scaling factors per window for each studied gas: XCO$_2$, XCH$_4$ and XCO at Peterhof.**

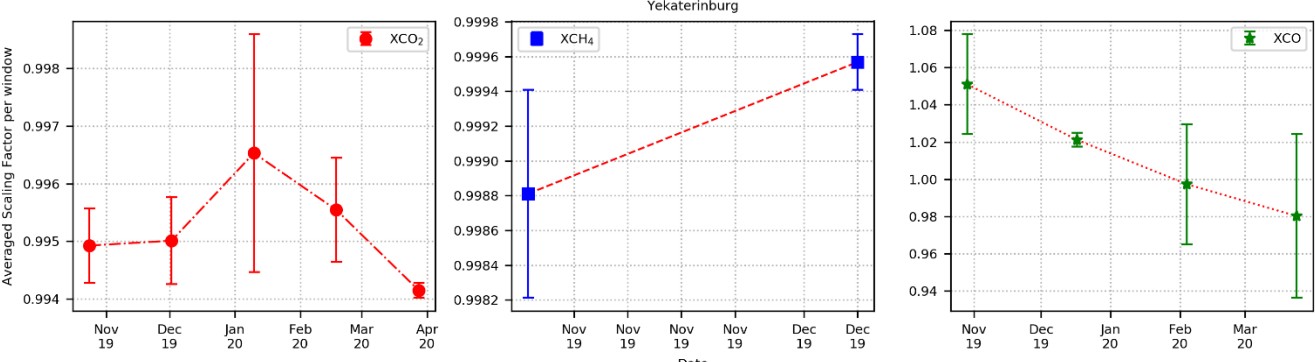

**Figure A- 11 As Figure A-8 but for Yekaterinburg.**



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
