# Peer review of "Investigation of space-borne trace gas products over St. Petersburg and Yekaterinburg, Russia by using COCCON observations"

_Atmospheric Measurement Techniques, 2021_

## Referee Comment (RC3)

This paper presents extensive comparison between satellite measurements and COCCON measurements at two high-latitude Russian cities, St. Petersburg and Yekaterinburg. A method of scaling CAMS model data to COCOON observations is developed, for a better comparison with the satellite measurements. I have several main concerns, which should be addressed before this paper can be published in AMT.

1. When comparing the satellite products with COCCON measurements, have you considered different averaging kernels for the satellite data and the ground-based remote sensing measurements?

2. For the regression plots shown in this paper, how are the R2 values determined? I understand that the fits are forced to go through origin. However, the reported R2 values are all very high, and I can not see how a R2 = 0.9999 is possible for the middle plot of Fig. 19, where there is no correlation between the scatter points and the regression line, and how in Figure 10 the top left and bottom left plots can have the same R2. Please check your regression algorithm.

3. You use a co-location criteria for satellite and ground-based measurements of up to 200 km. Have you checked whether there are emission sources in between? I doubt that the comparison can be objective if the distance is so large.

Further suggestions:
1. One map (maybe in appendix) regarding the locations of their measurements (up- an downwind sites) and the potential emission area which you assume to contribute the enhancement (details in Line 170, page 8). It is quite hard to imagine if someone is not familiar with the geographical information for your study.

2. Eq. 1: please explain "DT" and n

3. Eq. 2: please explain "t", which is usually referring the continuous time.

4. Line 538: "This discrepancy might be due to the COCCON observation during winter". So do you think COCCON measurements are not representative for the monthly mean?

5. All the legends: 'xCO2' should be big 'X'

6. In Figure.5, there are lots of solid vertical lines. Is there some special meaning regarding them?

7. In Figure.7, the information is quite hard to get. The dates are not readable from x-axis and also not equally distributed. If only showing the information that 22 days are available, maybe you can use a table to show the dates and some features of the measurements, e.g., daily mean +/- std. If the tendency is the key, clarify the x-axis and show the information clearly.

8. In Figure.8(b), the unit of XCH4 should be ppb instead of ppm;

9. In Line 285-289: how are these three collection radius chosen?

10. In line 300, does 'with short-term enhancement' mean those small fluctuations within one month before 2019-08? Please clarify it further.

11. For Figure 8 (b) XCH4, is there any explanation regarding the rising signals observed from all products in 2019 from summer to winter?

12. Please change the order of the figures in the appendix to follow the main paper content

13. Figure 16 and 17, XCO2 from CAMS-COCCON are bias-low compared to the values from GOSAT and OCO-2. It looks like a constant bias. Have you looked into the reason behind?

14. Line 210: you could consider to include the two following references mentioning the permanent network MUCCnet, which is a typical example of continuous deployment and a measurement campaign in US using COCCON spectrometers:
https://amt.copernicus.org/articles/14/1111/2021/
https://acp.copernicus.org/articles/21/13131/2021/acp-21-13131-2021.html

15. Figure22(e), 'Delta XO' in x-axis should be 'Delta XCO'

16. Line 37: here is the first time when the abbreviation 'GHG' appears. The full name of GHG should be explained here, instead of the next line. Additionally, the information demonstrated in Line 37-38 (two sentences) is somehow repeated. Could you rewrite it?

17. Line 45: "on that regard" → "in this/that regard". Additionally, a comma should be added.

18. Line 48: "in 2005" instead of "on 2005"

19. Line 70: "column" instead of "columnar"?

20. Line 226: Schneider et al., (2020) instead of Schneider et al., 2020

21. Line 459: "showing RMSD as a function"

22. Please check your reference list. Some of the references are missing there.

---

## Author Comment (AC1)

We thank anonymous referee #1 for evaluating our manuscript and for the very useful comments, which we treat in the following item-by-item. In this author comments all the points one-by-one raised by the reviewer are replicated in blue text, along with the corresponding reply from the authors in black text.

L42: a reference would be good.

You are right, this would be appropriate. We have added the following references:

Hoegh-Guldberg, O., D. Jacob, M. Taylor, M. Bindi, S. Brown, I. Camilloni, A. Diedhiou, R. Djalante, K.L. Ebi, F. Engelbrecht, J. Guiot, Y. Hijioka, S. Mehrotra, A. Payne, S.I. Seneviratne, A. Thomas, R. Warren, and G. Zhou, 2018: Impacts of 1.5°C Global Warming on Natural and Human Systems. In: Global Warming of 1.5°C. An IPCC Special Report on the impacts of global warming of 1.5°C above pre-industrial levels and related global greenhouse gas emission pathways, in the context of strengthening the global response to the threat of climate change, sustainable development, and efforts to eradicate poverty [Masson-Delmotte, V., P. Zhai, H.-O. Pörtner, D. Roberts, J. Skea, P.R. Shukla, A. Pirani, W. Moufouma-Okia, C. Péan, R. Pidcock, S. Connors, J.B.R. Matthews, Y. Chen, X. Zhou, M.I. Gomis, E. Lonnoy, T. Maycock, M. Tignor, and T. Waterfield (eds.)]. In Press.

IPCC, 2021: Summary for Policymakers. In: Climate Change 2021: The Physical Science Basis. Contribution of Working Group I to the Sixth Assessment Report of the Intergovernmental Panel on Climate Change [Masson-Delmotte, V., P. Zhai, A. Pirani, S. L. Connors, C. Péan, S. Berger, N. Caud, Y. Chen, L. Goldfarb, M. I. Gomis, M. Huang, K. Leitzell, E. Lonnoy, J.B.R. Matthews, T. K. Maycock, T. Waterfield, O. Yelekçi, R. Yu and B. Zhou (eds.)]. Cambridge University Press. In Press.

L50: the reference is only 7 years after 2005, which is in conflict to the statement.

Correct, sorry for the incorrect statement. We have changed the sentence to:

Unfortunately, after several years the global anthropogenic emissions of GHGs continued increasing (Harris et al., 2012).

L59: 'long lived' should be specified.

We added the following information and reference:

[$CO_2$ …] is long lived because it has an atmospheric lifetime which spans from centuries to millennia (IPCC, 2018)

IPCC (2018). Summary for Policymakers. in Global Warming of 1.5 C: An IPCC special report on the impacts of global warming of 1.5 C above pre-industrial levels and related global greenhouse gas emission pathways, in the context of strengthening the global
response to the threat of climate change, [Masson-Delmotte, V., et al. (eds.)] 32 pp, World Meteorological Organization, Geneva, Switzerland.

L61: 'more crucial than ever' should be rewritten.

Thanks, we rephrased the sentence as follows:

Both applications require measuring relatively small changes over a large background concentration, which requires high-accuracy instrumentation and calls for continuous efforts on improving the instrumental and data processing state-of-the-art (Frey et al., 2019, Alberti et al., 2021).

L66: I know there is no freely available data from Tansat so far, but shouldn't it still be mentioned ?

Agreed, we have added this information and reference:

…, and the Chinese Carbon Dioxide Observation Satellite (TanSat) (Liu et al., 2018).

Liu, Y., Wang, J., Yao, L., Chen, X., Cai, Z., Yang, D., Yin, Z., Gu, S., Tian, L., Lu, N., and Lyu, D.: The TanSat mission:434 preliminary global observations, Sci. Bull., 63, 1200–1207, https://doi.org/10.1016/j.scib.2018.08.004, 2018.

L105: this region has 'high uncertainties' because of the high emissions and high fluxes. This then raises the question if the scaled COCCON product can contribute to lower these high uncertainties also on the global scale?

This is a very interesting question! As COCCON is on its way of becoming a quasi-global network, we expect that COCCON will be able to contribute to lower the uncertainties on global scale as well. This could be achieved by either generating "nudged" CAMS fields (which is what we did for a target region in the framework of our study) or by more advanced procedures of data assimilation. The usefulness of TCCON for this purpose has been demonstrated by Chevallier et al., 2011.

Chevallier, F., et al. (2011), Global $CO_2$ fluxes inferred from surface air-sample measurements and from TCCON retrievals of the $CO_2$ total column, Geophys. Res. Lett., 38, L24810, doi:10.1029/2011GL049899.

L315: There are very few (or no?) overpasses within 50km for Yekaterinburg. Do the points with a higher difference in Fig. 10 (d and e) correspond to the overpasses with a higher distance? Maybe a plot delta xGas vs. distance would clearify this (in the appendix).

This is a valid point. We have added the following paragraph in the paper and added the following Table 1 and the Figure 1 to the paper appendix.

The Table A-2 lists the number of coincidences (pixel-wise) for 50 and 100 km radius, and the number of coincident satellite pixels is reduced by a factor of 3 to 5 for the narrower radius. From the Figure A-5, we do see a tendency of slightly reduced differences with closer colocation within the 100 km limit in case of $CH_4$, but no clear tendency for the other species. Due to the low number of coincident measurements when using 50 km, we decided to accept the 100 km distance criterion for the Yekaterinburg observations.

**Table 1. Number of TROPOMI measurements within 50 km and within 100km, respectively.**

|          | R = 50 km | R = 100 km |
|----------|-----------|------------|
| $XCH_4$  | 101       | 345        |
| XCO      | 265       | 1111       |
| $XH_2O$  | 19        | 136        |

[Figure]

**Figure 1. Difference between a single TROPOMI measurement with the averaged COCCON measurement (±1 h of satellite overpass) with respect to their distance.**

---

## Author Comment (AC2)

Alberti et al. compared multiple products of total column measurements of greenhouse gases, including ground- and space-based observations and model simulations, at two high-latitude Russian cities, St. Petersburg and Yekaterinburg. As high-latitude total column observations are sparse, such evaluations are quite useful. The paper is well written, with some structural improvements, it will be suitable for publication at AMT.

General comments:

We would like to thank anonymous Referee #2 for evaluating our manuscript and for the constructive comments that will definitely improve it. Below, we list the original comments/questions in blue colour and our respective answers in black, respectively.

1. Many products are compared. However, it is not clear whether there is a reference. On the one hand, COCCON retrievals were described biased low compared to other products, on the other hand, COCCON retrievals were used to scale the CAMS simulations. Are COCCON retrievals linked to the WMO scales? In a similar way as done for TCCON. With the current version, it gives the readers an impression that various products were compared.

We believe that the COCCON data products can be regarded as to be linked to WMO scale and can be used as a reference for validation purposes. Firstly, strong emphasis is put on achieving an accurate calibration of each spectrometer using laboratory characterisation procedures and side-by-side solar measurements with the TCCON station in Karlsruhe and the COCCON reference spectrometer operated by KIT. The procedures are described by Frey et al., 2019, and Alberti et al., 2021. Secondly, for the COCCON network in total, the Xgas products generated with the PROFFAST code are calibrated using TCCON as the reference. TCCON in turn is linked to the WMO scale by using in situ-profiles. That the applied procedures are successful and has been investigated in the framework of the ESA project FRM4GHG. The results of this project have been published by Sha et al., 2019. Therefore, we do not believe that COCCON suffers from a low bias and we decided to scale the CAMS fields as suggested by the COCCON results. We regard the COCCON scale as reference in this comparison. We therefore state in the abstract that "These adjusted CAMS data are then used for satellite validation".

We agree that our statement "COCCON $XCO_2$ is biased low by about 0.8-3.1 ppm in comparison to CAMS and other satellite products" is misunderstood in this context. We therefore changed it into "CAMS and the satellite products show a high bias of about 0.81-3.1 with respect to COCCON."

2. The scaling of CAMS data based on the COCCON data is a practical way of obtaining more matches. The scaled CAMS data could be called bias corrected CAMS data, instead of upscaled COCCON data. Since COCCON retrievals are obtained by scaling the priors, if the priors would be CAMS, it will be more straightforward to obtain the scaling factor, correct?

We do not fully agree with the suggested wording of "bias corrected CAMS data", as we do not apply a global bias correction to the CAMS data, but a scaling which is variable on time scales of several weeks (see Fig A-5 to A-7 showing the variability of the scaling factor). We agree that the operation also removes a general bias between CAMS and COCCON (and implicitly handles COCCON as the true scale), but it also reduces e.g. seasonal variations that are imperfectly reproduced by the model. We therefore would like to maintain the wording concerning the generation of a "scaled" CAMS dataset (one also might think of instead using the term "adjusted" or "tuned" for the resulting CAMS dataset?).

In general, COCCON aims at delivering data, which are compatible with TCCON. For this reason, we decided to adopt the a-priori profiles assumed by TCCON for the COCCON data analysis for the manuscript in discussion. However, we agree with the referee that in the context of adjusting the CAMS model using the CAMS profile as a-priori choice would be preferable. Therefore, we have re-processed the COCCON dataset by using CAMS profiles as a-priori and we have updated all the figures and results accordingly.

Some detailed comments:

L84: in this region instead of on this region

Changed accordingly

L94: compare already means intercompare/inter-compare, just use compare

Changed accordingly

L224: please rephrase the sentence. The dry air column from the ECMWF simulations?

Sorry, our statement was confusing. We rephrased the sentence as follows:

XCO is computed by dividing the CO total column by the dry air column extracted from the co-located TROPOMI $CH_4$ file. This dry air column is obtained from the surface pressure and water vapour column as provided by the European Center for Medium-Range Weather Forecast (ECMWF) analysis (Schneising et al., 2019; Lorente et al., 2021)

L279: were averaging kernels considered in the integrating process?

In the revised version of the manuscript, the CAMS profiles are used as a priori for COCCON. Therefore, no smoothing correction appears in this profile.

L290: why is COCCON XCO2 biased low by about 0.81 – 3.1 ppm? Is the difference indeed caused by a bias in COCCON XCO2? How is it known?

We changed the wording accordingly. We are not aware of a COCCON bias.

L417: how many points? It seems that very little data is available at Yekaterinburg.

As mentioned in L190, for the whole period of measurements, a total of twenty measurements days were collected. Considering that the active measurement period was October-April (autumn-spring), which are not the best months in terms of sunny conditions, the amount of measurements is still sufficient for applying the scaling method.

L432: I wonder whether the linear regressions are significant? What are the R-squared values?

As we apply a scaling on the CAMS data, the required factor is deduced from a linear regression forced through the origin. Therefore, the resulting R value is very near to one.

L478: please show some objective ways of assessing the agreement as "close agreement" cannot be judged.

Thank you for pointing this out! We accordingly added the table below in the appendix, which contains the actual variability (standard deviation) over the full measurement period as indicated by CAMS, and the bias and standard deviation of the difference between CAMS and COCCON, and between scaled CAMS and COCCON for each studied city, respectively. The value in the column "CAMS variability" can be regarded as minimum requirement for an acceptable agreement; while a "close agreement" between adjusted model and observation should be a fraction of this value. As can be seen from the table, the standard deviation between scaled CAMS and COCCON is significantly smaller than the actual variability, so this justifies the statement of a "close agreement" and this applies for both cities.

We added the following statement to the text of the paper.

From the Table A-2 in the appendix, it can be observed that the bias and the standard deviation between scaled CAMS and COCCON is significantly smaller than the CAMS variability of the original data-set. This further demonstrates the "close agreement" between adjusted model and observation.

**Table A- 1. The variability (standard deviation) of the original CAMS products during the COCCON measurement period in each city, and bias and standard deviation for the difference between CAMS and COCCON, and between scaled CAMS and COCCON.**

| Species | Peterhof | | | Yekaterinburg | | |
|---|---|---|---|---|---|---|
| | Variability of original CAMS products | CAMS - COCCON | scaled CAMS - COCCON | Variability of original CAMS products | CAMS - COCCON | scaled CAMS - COCCON |
| $XCO_2$ | 3.45 ppm | $1.76 \pm 0.82$ ppm | $0.18 \pm 0.79$ ppm | 2.24 ppm | $1.31 \pm 0.69$ ppm | $-0.008 \pm 0.56$ ppm |
| $XCH_4$ | 11.81 ppb | $14.97 \pm 8.7$ ppb | $-1.95 \pm 6.84$ ppb | 5.95 ppb | $19.9 \pm 5.88$ ppb | $-0.58 \pm 4.19$ ppb |
| XCO | 10.67 ppb | $0.59 \pm 6.51$ ppb | $-1.92 \pm 4.90$ ppb | 11.58 ppb | $1.96 \pm 6.50$ ppb | $2.16 \pm 5.03$ ppb |

---

## Author Comment (AC3)

This paper presents extensive comparison between satellite measurements and COCCON measurements at two high-latitude Russian cities, St. Petersburg and Yekaterinburg. A method of scaling CAMS model data to COCOON observations is developed, for a better comparison with the satellite measurements. I have several main concerns, which should be addressed before this paper can be published in AMT.

We would like to thank anonymous Referee #3 for evaluating our manuscript and for constructive comments that will definitely improve it. Below, we list the original comments/questions in blue colour and our answers/comments in black, respectively.

1. When comparing the satellite products with COCCON measurements, have you considered different averaging kernels for the satellite data and the ground-based remote sensing measurements?

We would like to thanks our referee for pointing this out; in the initial manuscript, we did not use averaging kernels in both comparisons COCCON and CAMS-COCCON vs Available Satellite products, because we assumed that the adjustments resulting from the smoothing would be minor. Nevertheless, we agree that performing a comparison without removing the smoothing error arising from use of different a-priori profiles is a technical error and we therefore have updated the manuscript and applied the averaging kernels for removing the smoothing error bias from our comparisons. We added the section 4.2 as follows:

4.2 Removal of the smoothing error bias

Because we aim at comparing different data products from space-borne with COCCON products and each of them have different sensitivities and use a different a-priori profiles; it is important to account for these differences when comparing a defined Xgas specie as described by Rodgers and Connor, (2003) and Connor et al., (2008). Such procedures have been applied in other similar studies (Hedelius et al., 2016, Yang Yang et al., 2020, M. K. Sha et al., 2021). In this study, we used the method described in Connor et al., (2008). We took as starting point the eq. (13), then the state vector can be written as:

$$\overrightarrow{VMR}_{gas,obs} = \overrightarrow{VMR}_{gas,apr} + A(\overrightarrow{VMR}_{true} - \overrightarrow{VMR}_{gas,apr}) \qquad \text{Eq. 1}$$

Where $\overrightarrow{VMR}$: represents the Volume Mixing Ratio. The left-term of the equation represent the retrieved value, while the right term represents the VMR calculated based on the a-priori plus the effect of the averaging kernel matrix A applied to difference of the VMR between the true atmospheric gas concentration and the a-priori. By dividing the atmosphere in "k" layers, this equation can be written as follows:

$$X_{gas,obs} = X_{gas,apr} + \sum_0^k h_k a_k (VMR_{true,k} - VMR_{apr,k}) \qquad \text{Eq. 2}$$

Where:
$X_{gas,y} = \sum_k h_k . VMR_{y,k}$ With "y" being a defined a-priori used and $h_k$: the pressure-weighting function in a defined layer "k" (Connor et al., 2008), i.e:

$$h_k = \frac{(p_{k-1} - p_k)}{p_0} \qquad \text{Eq. 3}$$

By using Eq. 2 with a "new" and "old" satellite-a-priori we obtain (*) and (**) as follows:

$$X_{gas,obs-new} = X_{gas,apr-new} + \sum_0^k h_k a_k (VMR_{true,k} - VMR_{apr-new,k}) \qquad (*)$$

$$X_{gas,obs-sat} = X_{gas,apr-sat} + \sum_0^k h_k a_k (VMR_{true,k} - VMR_{apr-sat,k}) \qquad (**)$$

Then we subtract (*) from (**):

$$X_{gas,obs-new} = X_{gas,obs-sat} + \left(X_{gas,apr-new} - X_{gas,apr-sat}\right)$$
$$+ \sum_0^k h_k a_k VMR_{true,k} - \sum_0^k h_k a_k VMR_{apr-new,k}$$
$$- \sum_0^k h_k a_k VMR_{true,k} + \sum_0^k h_k a_k VMR_{apr-sat,k}$$

Which turns into:

$$X_{gas,obs-new} = X_{gas,obs-sat} + \left(X_{gas,apr-new} - X_{gas,apr-sat}\right)$$
$$+ \sum_0^k h_k a_k \left(VMR_{apr-sat,k} - VMR_{apr-new,k}\right)$$

Where $X_{gas,obs-new}$ in Eq. 4 becomes the smoothed satellite product, which takes into account the a-priori used for the COCCON retrievals.

For using Eq. 4, both a-priori profiles need to be resampled on the same pressure grid. The vertical profiles used for the COCCON analysis are interpolated to the pressure levels of different satellite products (TROPOMI CO, GOSAT $CO_2$ and $CH_4$, OCO-2 $CO_2$ and OCO-2 FOCAL $CO_2$) by using the mass conservation method described in Langerock et al., (2015).

The smoothing correction is not applied to the $XH_2O$, because the natural variability of $XH_2O$ is very high anyway.

2. For the regression plots shown in this paper, how are the R2 values determined? I understand that the fits are forced to go through origin. However, the reported R2 values are all very high, and I can not see how a R2 = 0.9999 is possible for the middle plot of Fig. 19, where there is no correlation between the scatter points and the regression line, and how in Figure 10 the top left and bottom left plots can have the same R2. Please check your regression algorithm.

We would like to thanks to the referee for pointing this out. We agree that this can confuse the reader, when the fitting lines are forced to cross the origin point. We then decided to change to a linear regression without forcing to cross the origin point and therefore we have updated all the figures accordingly.

3. You use a co-location criteria for satellite and ground-based measurements of up to 200 km. Have you checked whether there are emission sources in between? I doubt that the comparison can be objective if the distance is so large.

We use a collocation radius of 200 km for the OCO-2 comparison because there are so few OCO-2 observations within 50 km or 100 km, see Table 1. We have added Figure 2 and Figure 3 below (on reply to question 9.) which indicates that there is no significant increase of bias when increasing the collocation radius.

**Table 1. Number of observations for OCO-2 and OCO-2 FOCAL within different collection radius at Peterhof and Yekaterinburg.**

| | OCO-2 | | OCO-2 FOCAL | |
|---|---|---|---|---|
| | Peterhof | Yekaterinburg | Peterhof | Yekaterinburg |
| 50 km | 5 | 1 | 0 | 0 |
| 100 km | 13 | 1 | 1 | 0 |
| 200 km | 23 | 5 | 13 | 0 |

Further suggestions:
1. One map (maybe in appendix) regarding the locations of their measurements (up- and downwind sites) and the potential emission area which you assume to contribute the enhancement (details in Line 170, page 8). It is quite hard to imagine if someone is not familiar with the geographical information for your study.

The requested map is already included in Figure A-2 (a) in the appendix of the paper, together with a HYSPLIT map with backward trajectories arriving to Peterhof on that day. However, to make this clear we have improved the map as suggested by adding a reference to the figure in the main text, and marking the name of the potential source into the appendix figure (see Figure 1 ).

[Figure]

**Figure 1** **Spatial distribution of CO emissions (tons/0.1 degree × 0.1 degree /year) from Sector-Specific Gridmaps: Combustion for manufacturing. Data source: EDGAR v5.0, 2015 (https://edgar.jrc.ec.europa.eu/dataset_ap50, last access: 04 August 2021); the map was generated with python basemap toolkit by using ArcGIS from a World Shaded Relief Model.**

2. Eq. 1: please explain "DT" and n

"n" is defined in L426 of the original manuscript. While for "DT" we have included the next sentence (in red)

These sub-windows have the characteristics of being non-overlapping and they form equally sized bins on the time axis, as defined in the Eq. 1(Eq. 5 in revised version), where 'DT' stands for Date-Time, which goes from the first to the last point of measurements.

3. Eq. 2: please explain "t", which is usually referring the continuous time.

Thank for pointing this out. We have changed "t" for "k" in Eq. 2(Eq. 6 in revised version) to avoid confusions with "continuous time" as you mentioned it, and additionally we have added the next sentence (in red).

The Root-Mean-Square-Deviation (RMSD), which is calculated with the Eq. 2(Eq. 6 in revised version), where "k" stands for the number of points considered during the scaling in each sub-window, between COCCON and the CAMS-COCCON data must be the lowest possible.

4. Line 538: "This discrepancy might be due to the COCCON observation during winter". So do you think COCCON measurements are not representative for the monthly mean?

We assume that the COCCON measurements in Yekaterinburg are less representative during winter in Yekaterinburg, because the number of observations is so limited. We rephrased the sentence in the revised manuscript as follows:

This discrepancy might be due to the limited number of COCCON observations during winter in Yekaterinburg (Only 12 days of measurements from November to Mach were available).

5. All the legends: 'xCO2' should be big 'X'

The suggested changes were made on Figures: 1, 2, 3, 5 and 7

6. In Figure.5, there are lots of solid vertical lines. Is there some special meaning regarding them?

Sorry for this mistake! The Figure had been adjusted by deleting the solid vertical lines, which is a problem of the resolution of the plot.

7. In Figure.7, the information is quite hard to get. The dates are not readable from x-axis and also not equally distributed. If only showing the information that 22 days are available, maybe you can use a table to show the dates and some features of the measurements, e.g., daily mean +/- std. If the tendency is the key, clarify the x-axis and show the information clearly.

Sorry for this mistake again! The Figure was adjusted accordingly.

8. In Figure.8(b), the unit of XCH4 should be ppb instead of ppm;

Thanks for this observation! The Figure has been modified accordingly.

9. In Line 285-289: how are these three collection radius chosen?

The bias between each TROPOMI $XCH_4$ observation and the mean value of COCCON observations 1h before and after satellite overpass time shows an increasing tendency with their distance (see Figure 1 below). The biases become more obvious when the distance is larger than 50 km. This situation is not obvious for XCO and $XH_2O$. To reduce the biases, we thus, use a collection radius of 50 km for TROPOMI at Peterhof.

However, less coincident data are found in Yekaterinburg due to shorter measurement period (See Figure 3). Due to this reason, we use 100 km for TROPOMI and GOSAT in Yekaterinburg.

As explained above (see **Table *1***), a 200 km collection radius is more appropriate for OCO-2 and OCO-2 FOCAL products, as only a few point for 100 km and up to 5 points for 50 km.

[Figure]

**Figure 2 Difference between a single TROPOMI measurement with the averaged COCCON measurement (±1 h of satellite overpass) with respect to their distance.**

[Figure]

**Figure 3 Same as Figure 2 but for Yekaterinburg.**

10. In line 300, does 'with short-term enhancement' mean those small fluctuations within one month before 2019-08? Please clarify it further.

We refer to the temporary events of enhanced $XCH_4$, with a typical duration of a week. These might be connected to synoptic processes (variable tropopause altitude), which reside on the same time scale. We have added in the revised version "short-term enhancements of about a week duration, probably related to synoptic variations"

11. For Figure 8 (b) XCH4, is there any explanation regarding the rising signals observed from all products in 2019 from summer to winter?

These seasonal variabilities are of geophysical origin (variation of sources and sinks, atmospheric lifetime, etc.) and are well produced both by the CAMS model and the observations.

12. Please change the order of the figures in the appendix to follow the main paper content

Thanks for this observation! The figures have been sorted accordingly.

13. Figure 16 and 17, XCO2 from CAMS-COCCON are bias-low compared to the values from GOSAT and OCO-2. It looks like a constant bias. Have you looked into the reason behind?

The calibration of the COCCON XCO$_2$ product will be re-examined in the framework of the just started ESA project FRM4GHG-2. However, we have found excellent agreement of the COCCON XCO$_2$ calibration with boreal TCCON sites with the current calibration (Sha et al., 2021). The observed boreal sites are low-albedo targets from the satellite perspective; one might speculate that this causes a bias in the satellite observations. Further COCCON observations should be collected in future at these boreal sites.

14. Line 210: you could consider to include the two following references mentioning the permanent network MUCCnet, which is a typical example of continuous deployment and a measurement campaign in US using COCCON spectrometers:

https://amt.copernicus.org/articles/14/1111/2021/
https://acp.copernicus.org/articles/21/13131/2021/acp-21-13131-2021.html

We considered appropriate and therefore we have included these two references to the paper.

15. Figure22(e), 'Delta XO' in x-axis should be 'Delta XCO'

The Figure was modified accordingly and updated in the manuscript.

16. Line 37: here is the first time when the abbreviation 'GHG' appears. The full name of GHG should be explained here, instead of the next line. Additionally, the information demonstrated in Line 37-38 (two sentences) is somehow repeated. Could you rewrite it?

Thanks for these observations! The full sentence between Line 37-39 has been written again, keeping GHGs once and deleting the repetition. The added new-sentence is the following:

Global warming is one of the most discussed negative effects caused by the anthropogenic emissions of greenhouse gases (GHGs); mainly carbon dioxide (CO$_2$), methane (CH$_4$), and nitrous oxide (N$_2$O).

17. Line 45: "on that regard" => "in this/that regard". Additionally, a comma should be added.

Changed accordingly.

18. Line 48: "in 2005" instead of "on 2005"

Changed accordingly

19. Line 70: "column" instead of "columnar"?

Changed accordingly

20. Line 226: Schneider et al., (2020) instead of Schneider et al., 2020

Changed accordingly

21. Line 459: "showing RMSD as a function"

Changed accordingly: "the" deleted

22. Please check your reference list. Some of the references are missing there.

We have carefully checked the reference list and added the missing references and the additions after the review process suggested for the three anonymous referees.

References

Connor, B. J., Boesch, H., Toon, G., Sen, B., Miller, C., and Crisp, D. (2008), Orbiting Carbon Observatory: Inverse method and prospective error analysis, J. Geophys. Res., 113, D05305, doi:10.1029/2006JD008336.

Dietrich, F., Chen, J., Voggenreiter, B., Aigner, P., Nachtigall, N., and Reger, B.: MUCCnet: Munich Urban Carbon Column network, Atmos. Meas. Tech., 14, 1111–1126, https://doi.org/10.5194/amt-14-1111-2021, 2021.

Hedelius, J. K., Viatte, C., Wunch, D., Roehl, C. M., Toon, G. C., Chen, J., Jones, T., Wofsy, S. C., Franklin, J. E., Parker, H., Dubey, M. K., and Wennberg, P. O.: Assessment of errors and biases in retrievals of XCO2, XCH4, XCO, and XN2O from a 0.5 cm–1 resolution solar-viewing spectrometer, Atmos. Meas. Tech., 9, 3527–3546, https://doi.org/10.5194/amt-9-3527-2016, 2016.

Jones, T. S., Franklin, J. E., Chen, J., Dietrich, F., Hajny, K. D., Paetzold, J. C., Wenzel, A., Gately, C., Gottlieb, E., Parker, H., Dubey, M., Hase, F., Shepson, P. B., Mielke, L. H., and Wofsy, S. C.: Assessing urban methane emissions using column-observing portable Fourier transform infrared (FTIR) spectrometers and a novel Bayesian inversion framework, Atmos. Chem. Phys., 21, 13131–13147, https://doi.org/10.5194/acp-21-13131-2021, 2021.

Langerock, B., De Mazière, M., Hendrick, F., Vigouroux, C., Desmet, F., Dils, B., and Niemeijer, S.: Description of algorithms for co-locating and comparing gridded model data with remote-sensing observations, Geosci. Model Dev., 8, 911–921, https://doi.org/10.5194/gmd-8-911-2015, 2015.

Rodgers, C. D., and Connor, B. J. (2003), Intercomparison of remote sounding instruments, J. Geophys. Res., 108, 4116, doi:10.1029/2002JD002299, D3.

Sha, M. K., Langerock, B., Blavier, J.-F. L., Blumenstock, T., Borsdorff, T., Buschmann, M., Dehn, A., De Mazière, M., Deutscher, N. M., Feist, D. G., García, O. E., Griffith, D. W. T., Grutter, M., Hannigan, J. W., Hase, F., Heikkinen, P., Hermans, C., Iraci, L. T., Jeseck, P., Jones, N., Kivi, R., Kumps, N., Landgraf, J., Lorente, A., Mahieu, E., Makarova, M. V., Mellqvist, J., Metzger, J.-M., Morino, I., Nagahama, T., Notholt, J., Ohyama, H., Ortega, I., Palm, M., Petri, C., Pollard, D. F., Rettinger, M., Robinson, J., Roche, S., Roehl, C. M., Röhling, A. N., Rousogenous, C., Schneider, M., Shiomi, K., Smale, D., Stremme, W., Strong, K., Sussmann, R., Té, Y., Uchino, O., Velazco, V. A., Vigouroux, C., Vrekoussis, M., Wang, P., Warneke, T., Wizenberg, T., Wunch, D., Yamanouchi, S., Yang, Y., and Zhou, M.: Validation of methane and carbon monoxide from Sentinel-5 Precursor using TCCON and NDACC-IRWG stations, Atmos. Meas. Tech., 14, 6249–6304, https://doi.org/10.5194/amt-14-6249-2021, 2021.

Yang, Y., Zhou, M., Langerock, B., Sha, M. K., Hermans, C., Wang, T., Ji, D., Vigouroux, C., Kumps, N., Wang, G., De Mazière, M., and Wang, P.: New ground-based Fourier-transform near-infrared solar absorption measurements of XCO2, XCH4 and XCO at Xianghe, China, Earth Syst. Sci. Data, 12, 1679–1696, https://doi.org/10.5194/essd-12-1679-2020, 2020.